# Progress in Polycrystalline SiC Growth by Low Pressure Chemical Vapor Deposition and Material Characterization

**DOI:** 10.3390/mi16030276

**Published:** 2025-02-27

**Authors:** Michail Gavalas, Yann Gallou, Didier Chaussende, Elisabeth Blanquet, Frédéric Mercier, Konstantinos Zekentes

**Affiliations:** 1SIMAP, Grenoble INP, University Grenoble Alpes CNRS, 38402 Grenoble, France; didier.chaussende@grenoble-inp.fr (D.C.); elisabeth.blanquet@simap.grenoble-inp.fr (E.B.); frederic.mercier@grenoble-inp.fr (F.M.); 2Microelectronics Research Group, IESL/FORTH & University of Crete, 71003 Heraklion, Greece; 3MERSEN, 41 rue Jean Jaurès, 92230 Gennevilliers, France; yann.gallou@grenoble-inp.fr; 4CROMA, Grenoble INP, University Grenoble Alpes CNRS, 38000 Grenoble, France

**Keywords:** LPCVD, polycrystalline material, thin films, SiC, material characterization

## Abstract

The purpose of this paper is to give a review on the state of the art of polycrystalline SiC material grown by low-pressure chemical vapor deposition (LPCVD). Nowadays, LPCVD is the main technique used for the deposition of polycrystalline SiC, both in academic research and industry. Indeed, the LPCVD technique is today the most mature technique to grow high purity polycrystalline thin films with controlled thickness and structure over a large area (>50 cm) and/or 3D substrate. Its ability to have a high degree of modification on the growth conditions and the chosen precursor system allows the deposition of polycrystalline SiC films in various substrates with tailored properties according to the desired application. After a short introduction on the SiC material and its growth by the LPCVD technique, a review of theoretical studies (thermodynamics and kinetics) related to the CVD SiC growth process is given. A synthesis of the experimental studies is made focusing on the effect of the growth conditions on the properties of the deposited SiC polycrystalline material. Despite the numerous results, a full understanding of them is limited due to the complexity of the LPCVD process and the polycrystalline SiC structure. The conclusions show that the growth conditions, like temperature, chamber pressure, (C/Si)_(g)_, (Cl/Si)_(g)_, and doping have an impact on the microstructure and on the corresponding properties of the polycrystalline SiC films. Future perspectives are given in order to improve our understanding on the polycrystalline–SiC–LPCVD process and to enable the growth of tailor-made polycrystalline SiC films for future applications.

## 1. Introduction

Silicon carbide (SiC) is a material with exceptional mechanical, electrical, and thermal properties, making it suitable for a wide range of applications. Among the various forms of SiC, polycrystalline SiC growth by low-pressure chemical vapor deposition (LPCVD) stands out due to its cost-effectiveness and suitability for large-area depositions. Several companies [1,2,3,4,5] employ polycrystalline SiC coatings due to its wear-resistant, high corrosion, and heat resistance properties, as well as its excellent thermal conductivity. Polycrystalline SiC is also used in heating elements [1,2]. Furthermore, conductive n-doped polycrystalline SiC substrates for the fabrication of electric power applications are also industrially developed [3,4,5]. The aim of this paper is to provide a comprehensive review of the state of the art of polycrystalline SiC material grown by LPCVD. The motivation for this review stems from the increasing demand for high-performance SiC material and the need for a deeper understanding of the LPCVD growth process and its impact on material properties.

After a short introduction on the SiC material and its growth by LPCVD, theoretical studies (thermodynamics and kinetics) related to the CVD SiC growth process are reviewed. Then, a synthesis of experimental studies, emphasizing the effect of growth conditions on the properties of the deposited SiC polycrystalline material is given. This part explores the influence of the process parameters, such as the temperature, pressure, precursor composition, and gas flow, on the structural, mechanical, and electrical properties of the SiC thin films. Finally, the paper discusses the effects of the process parameters on the electrical resistivity and the thermal conductivity of the material. By providing a detailed analysis of the LPCVD process and its influence on polycrystalline SiC properties, this paper seeks to serve as a valuable resource for researchers and engineers working in the field of polycrystalline SiC materials.

## 2. Polycrystalline SiC: Material, Synthesis Processes and Applications

### 2.1. Structure and Properties of Polycrystalline SiC Material

Silicon Carbide crystals are formed by bonded Si–C pairs through a sp^3^ hybridization, arranged in a tetrahedral configuration of Si_4_C or SiC_4_, whereby each Si and C atom has four neighbors. Unlike its single crystalline counterpart, polycrystalline SiC consists of crystallites of a finite size (from nm to μm), separated by the grain boundaries which often are of an amorphous nature. The grain boundaries play a crucial role in the microstructure of the polycrystalline SiC and its corresponding properties; this will be discussed later in this review.

#### 2.1.1. Polycrystalline SiC Form Versus the Epitaxial and Amorphous SiC

Silicon Carbide is known for its polytypism. Polytypism occurs when a material can have different crystal structures varying in one dimension (stacking sequence) without changing its chemical composition. Those structures are called polytypes, and SiC has more than 200 different polytypes [6]. It is shown that their relative stability and their nucleation probability depend on temperature and other factors, like supersaturation or doping atoms [6].

The structure of polycrystalline SiC grown by CVD consists usually of a different orientation 3C-SiC polytype, also called β-SiC. For this polytype, there is a bilayer stacking sequence of ABCABC between the Si and C atoms [7]. It has a unit cell with a cubic zinc blended structure and a lattice constant of approximately 4.36 Å (Table 1) [6,7]. Due to its cubic structure, the crystallographic Miller indices are three (hkl), and the corresponding crystallographic directions and planes are noted by those three indices. For polycrystalline 3C-SiC, the usual preferential crystallographic orientations are the (111), (110), and (211), with their corresponding crystallographic planes to be (111), (110), and (211) [8,9,10]. Cubic 3C-SiC is a polytype stable at low temperatures (below 1800 °C) that can be grown in high supersaturation conditions, such as those reached in CVD [11]. However, the cubic 3C-SiC polytype is generally not stable at temperatures higher than 1900 °C, and it transforms into hexagonal polytypes, like 4H-SiC or 6H-SiC. Thus, the final structure depends on the synthesis process and conditions. Amorphous SiC is obtained at low temperatures (mostly below 800 °C) [12].

Silicon Carbide is a stoichiometric compound semiconductor with 50% silicon (Si) and 50% carbon (C) [6]. Nevertheless, polycrystalline SiC can present an excess of Si and C material in addition to the stoichiometric SiC grains. Thus, the synthesis of polycrystalline SiC material by Si and C sources can result in SiC_(s)_, or SiC_(s)_ + Si_(s)_, or SiC_(s)_ + C_(s)_ material.

#### 2.1.2. Physical and Chemical Properties of SiC

Silicon Carbide has a density of 3.21 g/cm^3^ and a molecular weight of 40 g/mol. The energy band gap depends on the SiC polytype, varying between 2.3 eV and 3.2 eV, almost three times greater than that of Si (Table 1) [6,7,13]. The Si–C bond is very strong (~4.6 eV), which is responsible for many of its physical and chemical properties. For example, it is very resistive in acids and bases at room temperature [7,13,14]. Although SiC is not stable in oxidizing environments at high temperatures, it can form a protective oxide scale (SiO_2_) on the surface of the SiC, effectively limiting further oxidation. This protection remains effective up to the melting point of SiO_2_ (1723 °C) [14]. Moreover, SiC can resist the hydroxides and molten basic salts attack up to 450 °C [14,15].
micromachines-16-00276-t001_Table 1Table 1Basic properties of SiC polytypes at 300 K (given by [6,7,13,16]).Polytype3C-SiC4H-SiC6H-SiCStructureCubicHexagonal HexagonalEnergy band gap (eV)2.363.263.02a-axis constant (Å)4.363.083.08c-axis constant (Å)-10.0815.11Average CTE (10^−6^ K^−1^) 4.52.302.25Intrinsic carrier concentration (cm^−3^)0.13–0.155–8 · 10^−7^10^−6^Relative dielectric constant 9.7
9.7Breakdown electric field E_B_ (10^5^ V·cm)254032μ_n_ * (cm^2^V^−1^s^−1^)750800400μ_p_ * (cm^2^V^−1^s^−1^)4011590* For dopant values N_d_, Na 10^16^ cm^−3^.

#### 2.1.3. Mechanical Properties

Silicon Carbide has a hardness of 9.2–9.3 according to the Mohs scale, second only to diamonds [9,16,17]. Regarding its cubic polycrystalline form, the Young modulus has been reported several times with a variable value in the range of 380–700 GPa [17] and a Poisson ratio around 0.18 [18]. Polycrystalline 3C-SiC has an acoustic wave velocity of 12.6 km/s at room temperature [13]. It is also shown that SiC can retain its mechanical properties (hardness and elasticity) even at very high temperatures [6].

#### 2.1.4. Electrical Properties

Cubic SiC has a breakdown field as high as 2.4 MV/cm [7], which is ten times higher than that of Si. The intrinsic average electron mobility (μ_n_) is 400 cm^2^V^−1^s^−1^. Polycrystalline doped SiC n-type and p-type is possible with an electrical resistivity as low as 5 mΩ·cm and 10^6^ mΩ·cm for the n-type [19] and p-type [20] material, respectively.

#### 2.1.5. Thermal Properties

Silicon Carbide has high thermal conductivity, exceeding that of Si, GaN, and Al_2_O_3_, although its value can vary with the polytype, microstructure, and the doping type and concentration [6,13]. The highest room temperature thermal conductivity, at room temperature, for polycrystalline 3C-SiC grown by CVD has been experimentally measured around 250 W·m^−1^·K^−1^ at room temperature [21,22]. In addition, it has a low (4-to-6 × 10^−6^ °C^−1^) thermal expansion coefficient, depending on the studied temperature range [23] due to the strong covalent bond length of Si–C (1.88 Å) [24].

### 2.2. Applications of Polycrystalline SiC

**Power electronics, MEMs, solar cells**: N-doped polycrystalline SiC with very low electrical resistivity (~5 mΩ·cm [19]) is used to fabricate engineered substrates for the fabrication of power devices required in electric vehicles [4]. N-doped polycrystalline SiC is suitable as a structural layer for MEM devices [25,26,27], due to the combination of its low electrical resistivity and mechanical properties (low residual stress—<500 MPa [25,26] and low strain gradient (e.g., 5.8 · 10^−4^ [25]), as well as due to its ability to grow on various substrates (SiO_2_, Si, graphite, Al_2_O_3_). The higher Young modulus compared to Si (ranging from 420–600 GPa [27]) makes it attractive for micro-mechanical resonators [26]. Thanks to its potentially low electrical resistivity (30 mΩ·cm [28,29]) and thermal conductivity (18–65 W/mK [30]), along with its high thermal stability (up to 2000 °C [29]), n-doped polycrystalline SiC is an attractive material for energy conversion applications at high temperatures [28,29], as well as for photovoltaic applications [31,32,33,34]. Indeed, the high thermal conductivity reduces the need for cooling systems, therefore minimizing the device’s volume and overall cost. Note that, in what concerns high temperature operation, the ohmic contact behavior of an n-type polycrystalline SiC/TiW interface has been stable up to 2000 °C [35].**High temperature coatings:** Polycrystalline SiC is suitable for high temperature coatings, as it is stable at 2000 °C [30,36,37] and has an excellent thermal conductivity. Its mechanical strength (fracture strength of 23 GPa [37] and tensile strength of 1.4–4 GPa [30]) make it ideal for high temperature applications, like aerospace turbofans [36], automotive parts (e.g., engines [37] and brake pads), and nuclear reactor sensors [30].**Chemical resistance and radiation-resistance applications:** Due to its very stable chemical structure (Si–C bond), SiC is attractive for fabricating devices operating in harsh chemical environments (like space missions and corrosive chemical environment [14]). Polycrystalline 3C-SiC coatings show excellent acid resistance. For instance, SiC-coated quartz glass dipped in a mixed solution of 5% HF and 5% HCl at 80 °C showed almost 33% less weight loss compared to pure quartz glass after an 8 h acid test [38]. Silicon Carbide ceramic optical mirrors have been employed for space telescopic systems to operate at a few Kelvin temperatures under space radiation [39].**Supercapacitors:** Good electrical conductivity along with the tailorable structure and its chemical resilience, as well as its excellent cycling stability makes polycrystalline SiC suitable as active electrode material for supercapacitor applications (electric vehicles, energy storage applications, or even portable electronic devices [40,41,42,43]). Electrodes with a large electrochemical surface area are possible due to surface defects and a tunable pore structure, thus increasing the capacitive performance [42,44]. Furthermore, polycrystalline SiC has also been demonstrated as a supporting/seed material (for example, in the case of carbon layer growth) for the fabrication of electrodes by mixed composited materials [40,45,46].**Bio-applications:** The biocompatibility of SiC along with its excellent chemical stability and its electrical (resistivity of <30 mΩ·cm) and mechanical properties (residual stress < 500 MPa and surface roughness < 50 nm) make it a good choice for bio-applications [47] and especially for neural interfaces applications [10,48]. Nanocrystalline SiC has been tested as a potential long-term luminescent biological probe with photostability comparable to that of organic dye propidium (PI) and tunable light emission for multicolor imaging [49,50]. Surface functionalization and passivation of 3C-SiC by biological ligands and organic molecules is also promising for biotechnology applications [51]. Accelerated aging tests show that nitrogen-doped polycrystalline SiC is electrically and chemically stable (non-detectable etching) for more than 70 days in a heated (at 90 °C) buffer saline solution [10].**Tribological applications:** Silicon Carbide ceramics have a very low coefficient of friction (COF) in water (as low as 0.01 [52]) and they are capable to be used as hydraulic systems and future machines, replacing the metal/oil lubrication systems, since they are environmentally friendly and save energy [52]. Silicon Carbide ceramics are also used in unlubricated systems [53]. The tribological behavior of SiC ceramics, both under lubrication and un-lubrication, depends on the microstructure, mechanical properties, surface characteristics, secondary phases, and external factors. It has been shown that the tribological behavior is improved by modifying the microstructure: for example, by reducing the grain size and the percentage of the grain boundary/intergranular phases, by having elongated SiC grains instead of equiaxed grains, by crystallizing the amorphous grain boundary phases, by hardening the intergranular phases, or even by controlling the formation of cracks [52]. In addition, good self-lubrication properties for SiC ceramics can be obtained by heat treatment, adding a secondary phase or producing porous SiC ceramics, and by surface processing or surface modification [53]. Research in biotribology shows the potential of SiC ceramics, as the COF of polycrystalline SiC in blood plasma can be similar to that in water by modifying the surface texture and roughness of SiC used in artificial organs [54].**Water splitting:** He et al. [55] have demonstrated 3C-SiC nanocrystals with efficient catalytic properties for the production of hydrogen. It is reported that the electrochemical activity in the formation of –H and –OH groups is inversely proportional with the SiC crystallite size, hence the specific surface area which affects the autocatalysis. In this dynamic process, the sustained adsorption and dissociation of H_2_O molecules on 3C-SiC nanocrystals allows for efficient hydrogen production [56]. Theoretical studies have also investigated the mechanism of the dissociation of the Si–H and Si–OH in the ultra-small nanocrystalline SiC [57]. Another study reports the photocatalytic ability of the n-type polycrystalline SiC to produce H_2_ in various chemical solutions [58]. Porous SiC nanowires array for photocatalysis have also been demonstrated [59]. Polycrystalline SiC is promising for the efficient hydrogen production that is necessary for future hydrogen fuels [60].

### 2.3. Growth Methods of Polycrystalline SiC Films

**Sputtering:** Polycrystalline 3C-SiC films are grown with rf (radiofrequency) sputtering at substrate temperatures higher than 500 °C and chamber pressure of some mbar [61,62,63]. Silicon (100) or (111) is the frequent substrate of polycrystalline SiC sputtering growth. Both single-crystalline SiC [61,63] and amorphous SiC [62,63] targets have been used. The typical growth rate is in the order of several hundreds of nm/h [61,63]. It is shown that the crystallite size is proportional to the substrate’s temperature [62]. Deposition at room temperature resulted in the deposition of a-SiC films [61,62]. A study of DC (direct current) planar magnetron sputtering reported the growth of stoichiometric polycrystalline SiC films on Si substrates, using a gas mixture of argon (Ar) and methane (CH_4_), as well as an Si wafer target [64].**Thermal spray:** Polycrystalline SiC-6H coatings for tribology testing with good adhesion and mechanical stability have been produced by the thermal spray technique [65,66]. A mixing of polycrystalline SiC powder with YAG (yttrium aluminum garnet) particles is used for the hot spray process, leading to the formation of thick polycrystalline SiC coatings (10–100 μm) with low wear resistance [65]. It is shown that the spray flame power and solid content strongly affects the thickness and the quality of the deposited coating [66]. Recent studies show that the cold spray technique is also capable of producing polycrystalline SiC [67].**Pack cementation:** Thermogravimetric analysis (TGA) of cement mixtures SiC–SiO_2_ [68,69] showed the synthesis of polycrystalline 3C-SiC coatings on carbon substrates in the temperature range of 1400–1700 °C and atmospheric pressure of Ar_(g)_. It is shown that the reaction rate strongly depends on the furnace temperature with an Arrhenius relationship indicating that the process is limited by the surface kinetics of the diffused molecules from the cement powder to the substrate [68]. For a 2 μm thick carbon substrate, a 1 μm thick polycrystalline SiC layer is produced [69]. The produced layers were porous, with a grain size and a porous size of 75 nm and 25 nm, respectively [69].**APCVD:** Atmospheric pressure CVD (APCVD) was mostly used in the 1980s and 1990s for the growth of polycrystalline SiC films [70]. However, since the improvement of the LPCVD technology for polycrystalline SiC deposition, the APCVD is less used [71]. Reports have demonstrated the deposition of polycrystalline SiC films in atmospheric pressure on polysilicon, SiO_2_, and Si_3_N_4_ substrates [72,73]. A two-step deposition process of polycrystalline SiC on polysilicon substrates through a seed carbon layer has been demonstrated [72]. Nevertheless, the simultaneous injection of SiH_4_ and C_3_H_8_ at 1050 °C successfully grows stoichiometric polycrystalline SiC films on SiO_2_ and Si_3_N_4_ amorphous substrates where the carbon seed layer is not possible [73]. Methyltrichlorosilane (MTS) is used as a single-source (Si and C containing molecules) precursor for high quality, stoichiometric polycrystalline SiC films grown at 1000 °C [74], while, at higher temperatures, a single crystalline SiC is deposited. Growth rates for polycrystalline SiC is in the order of 1 to several μm/h. An injection of phosphine gaseous precursors is able to deposit n-doped polycrystalline films on SiO_2_ and Si_3_N_4_, respectively [73].**LPCVD:** Low-pressure CVD (LPCVD) is the most frequent (see Figure 1) technique for the deposition of polycrystalline SiC films. Both cold wall [27] and hot wall [25] reactors are used for polycrystalline SiC deposition. The LPCVD reactors enable the use of a variety of precursors, both single source and double precursors, and reduce the incorporation of impurities of the deposited film [75]. In situ doping (mostly n-type) is also possible by adding a doping agent during growth.
Figure 1Percentage of papers found in Web of Science and Google Scholar (from the 1960s to 2024) for polycrystalline SiC thin film growth methods (Keywords: polycrystalline SiC; thin films; coatings; LPCVD; APCVD; PECVD; sputtering; thermal spray; pack cementation). The absolute number of papers is also reported in parenthesis.
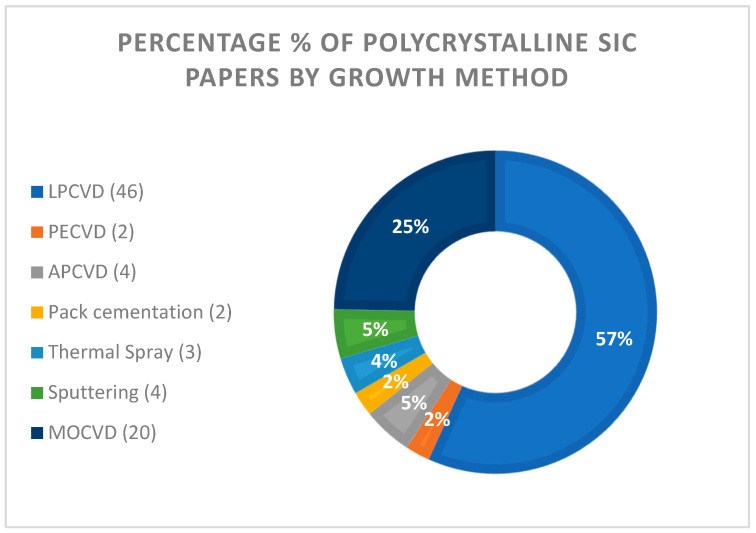


**PECVD:** This technique uses plasma for the decomposition of the CVD precursors along film growth and, thus, at a lower temperature than in thermal CVD. The lower temperature growth is the main advantage of the PECVD. Frequently double precursors systems, like SiH_4_ and CH_4_, are used for the Si and C source, respectively [34,76], while single source precursors (e.g., tetramethylsilane [42]) have been employed as well. In contrast to the amorphous counterpart (a-SiC), for which the PECVD is classically used, the state of the art for crystalline SiC growth by PECVD is poor. Zhuang et al. [42] has reported the deposition of polycrystalline SiC on single crystalline Si (100) using microwave plasma enhanced CVD (MWCVD). To our knowledge, doping of polycrystalline SiC grown by PECVD is not reported. Nitrogen doping of films in the literature is limited to a-SiC PECVD deposition [70].**Comparison between polycrystalline SiC film growth techniques:** Plenty of scientific work has been reported for the growth of polycrystalline SiC. The CVD technique has been developed for the growth of dense polycrystalline SiC films, and is the most common method for coatings in industry. The LPCVD technique is today the most mature technique to grow high purity polycrystalline thin films with controlled thickness and structure over a large area (>50 cm) and/or 3D substrates [77]. Compared to the APCVD, the vacuum in LPCVD allows for a better control of the deposition conditions, improving the film’s quality and the thickness uniformity [70]. The variation of sub-atmospheric pressure allows for a higher degree of modification on the growth conditions, the deposition process, and thus the corresponding properties of the produced film [70]. LPCVD can also operate in lower temperatures than APCVD, which makes it a less expensive choice for systematic research and the industrial production of polycrystalline SiC. With the PECVD technique, it is difficult to produce high quality polycrystalline SiC free from hydrogen and microvoids [12]. Thermal spray is ideal for producing polycrystalline SiC mixtures but, despite its high growth rate, it is difficult to obtain stoichiometric coatings. The cementation process has a small growth rate and the produced SiC coatings are compromised by a porous structure due to the thermally limited diffusion of Si species into the carbon substrate and the gaseous production of oxygen species.

### 2.4. LPCVD for Growing Polycrystalline SiC

Low-pressure CVD (LPCVD) is the main technique to grow polycrystalline 3C-SiC (simplified to “polycrystalline SiC” in the following). In general, the LPCVD reactors operate under low pressure, generally between 0.1 mbar and a few hundreds of mbars (mostly in the order of mbar), and low temperature, generally between 800 °C and 1500 °C. Decreasing the operation pressure can increase the gas diffusivity, thus enhancing the uniformity of the deposited thin film [27]. Amorphous or nanocrystalline SiC is usually deposited below 800 °C in the LPCVD technique [78]. Both cold wall and hot wall reactors are used for low pressure depositions. Many substrates can be used, like SiO_2_ [79], Si_3_N_4_ [80], or graphite [8,81,82]. Polycrystalline SiC is mainly synthesized from the C–H–Si system (organosilanes) or the C–Cl–H–Si (chlorinated organosilanes) system. Both single-source and double-precursor systems (two precursors, one Si containing species, and one C containing species) are equally used for the deposition of polycrystalline SiC thin films. Double precursor systems provide greater flexibility in terms of precursor ratios, allowing for higher customization of the material’s properties. Single-source precursor systems enable depositions at lower temperature conditions (often in a range of 700–1000 °C) due to lower required decomposition energy. Furthermore, the apparatus can be simplified with a single-source precursor. The precursors are traditionally diluted in H_2(g)_ (also acting as a reactant) and sometimes in Ar_(g)_ (inert). The addition of NH_3(g)_ and/or N_2(g)_ during growth allow the deposition of n-type polycrystalline SiC, while the addition of trimethylaluminium leads to p-type SiC [83,84].

### 2.5. Previous Review Papers on Polycrystalline SiC Films

Maboudian et al. [9] demonstrated the advances on the SiC thin films technology and the manufacturing technology of their low dimensional MEM devices. Despite the fact that the authors gave some basic properties of the polycrystalline SiC films grown by the LPCVD technique, the rest of the discussion is focused on the advances in the fabrication technology of SiC MEMs and the possible applications of SiC in microelectronic devices. Fraga et al. [70] reported the chemical vapor deposition techniques for SiC film deposition in general. In this review, a broad description of CVD and ALD growth methods for depositing epitaxial and polycrystalline SiC thin films is given. However, the basic elements of LPCVD (frequently used precursors, used substrates, and dopant agents) for growing polycrystalline SiC are mentioned briefly. Other works [85,86] have reported on the progress of several deposition techniques according to the deposited structure of SiC and its potential application, but the discussion for polycrystalline SiC is limited. Similarly, La Via et al. [18] summarized the emerging possible future applications, beyond the power electronics, of both hexagonal and cubic SiC films. The focus of this report is on the use of polycrystalline SiC in MEMs and bioapplications, without discussing the growth and the resulting physical properties. On the other hand, Li et al. [7] focused only on the prospects of cubic SiC for power electronic devices technology, and the discussion of the polycrystalline SiC properties is limited. Elgazzar and Elbashar [15] showed the differences between the CVD techniques and emphasized the pulsed laser deposition (PLD) technique on growing nanostructured and epitaxial SiC thin films along with the possible applications. Finally, Pedersen et al. [87] reported on a comprehensive discussion of the chloride-based CVD growth of epitaxial SiC films in comparison with Si films, emphasizing the understanding of the chlorine-based processes. Despite the variety of available review reports given for the synthesis of SiC material and its promising application, the information they give about the growth and the properties of polycrystalline SiC thin films is limited.

To our knowledge there is not a review paper that concentrates exclusively on polycrystalline SiC thin films deposition by the LPCVD technique and, more precisely, on the polycrystalline SiC microstructure and functional properties (electrical resistivity, thermal conductivity) as a function of the growth conditions as it is performed in the present paper.

## 3. LPCVD Growth of Polycrystalline SiC: Basic Elements

### 3.1. Dependence of SiC Film Chemical Composition on Growth Conditions—Theoretical Work

In the following discussion, the (C/Si)_(g)_ and (Cl/Si)_(g)_ ratios correspond to these element ratios in the precursors entering the growth chamber if there is no further precision. On the other hand, the (C/Si)_(s)_ ratio corresponds to the ratio of the two elements in the solid polycrystalline SiC film grown by the CVD process.

#### 3.1.1. Thermodynamic Modeling

**Growth Temperature effect:** Thermodynamic calculations in the system C–H–Si for different precursors, (SiH_4_–CH_4_–H_2_)_(g)_, (SiH_4_–C_2_H_4_–H_2_)_(g)_, and (SiH_4_–C_3_H_8_–H_2_)_(g)_ show very similar trends for the phase prediction [88,89]. Stoichiometric SiC deposition is possible from 1000 °C to 1500 °C for a small window of (C/Si)_(g)_ around 1 [88,89] (Figure 2a). This window is enlarged in very high temperatures (1600–1800 °C) for two reasons. On the one hand, above 1400 °C, Si_(s)_ is not thermodynamically stable and remains in the gas phase (melting point of Si = 1420 °C) thus reducing the possibility of Si_(s)_ cluster deposition. On the other hand, at a high temperature, the formation of CH_4(g)_ (a reaction product between the C and H_2_-carrier gas) reduces the C incorporation into the deposited film and thus increases the size of the pure SiC domain, something that explains the greater domain of stoichiometric SiC achieved with H_2(g)_ (solid lines in Figure 2a) carrier gas versus the one achieved with Ar_(g)_ (dotted lines in Figure 2a).

In the C–Cl–H–Si system, thermodynamics calculations have been performed for the following gas mixtures: (CH_3_SiCl_3_–H_2_)_(g)_ (methyltrichlorosilane) [89,90,91,92,93], (SiCl_4_–CCl_4_–H_2_)_(g)_ [89], (SiCl_4_–CH_4_–H_2_)_(g)_ [89,94], and (CH_3_)_2_SiCl_2_–H_2_)_(g)_ (dimethyldichlorosilane) [87]. All these studies agree that the two main solid phases are SiC_(s)_ and C_(s)_. Silicon co-deposition is almost never obtained and is only found in the (CH_3_SiCl_3_–H_2_)_(g)_ mixture for a very high dilution ratio ([H_2_]/[MTS] > 10^4^) [91,92]. The existence of a higher temperature domain in which SiC_(s)_ + Si_(s)_ is prevented is due to the presence of Cl that retains silicon in the gas phase, mainly in the form of SiCl_2(g)_ [88,93]. Also, it has been shown by different studies that an increase of temperature (above 1400–1600 °C) leads to the possible co-deposition of carbon [88,91,92,94]. An example is given on Figure 2b. Note that the same studies show the possible formation of C co-deposition also for temperatures below 800–1000 °C, suggesting that an optimal range of temperature to obtain the highest purity of SiC is between 1000 °C and 1400 °C.

**(C/Si)_(g)_ effect:** In the C–H–Si and C–Cl–H–Si systems, and for the temperature range of polycrystalline SiC (1000–1500 °C) growth, a co-deposition of Si_(s)_ or C_(s)_, is obtained for low and high (C/Si)_(g)_ values, respectively [88,90,94] (Figure 2a). In the case of the Si–C–H system, and for very high temperatures (T > 1500 °C), the (C/Si)_(g)_ has a lesser effect in the solid composition of SiC than for the C–Cl–H–Si system. However, experimentally there are very rare cases of C-rich films [95,96,97].

**(Cl/Si)_(g)_ effect**: The aim of using chlorine, in the case of epitaxial SiC growth, was to increase the growth rate while limiting the formation of Si droplets occurring at high flow rates, thus resulting in the reduction or even the elimination of SiC_(s)_ + Si_(s)_ material deposition [93,98]. Indeed, in non-chlorinated chemistry, the partial pressure of a Si precursor needs to be low to avoid the formation of free Si, but then the deposition rate is also slow. Conversely, chlorine, which has high affinity with Si, retains silicon in the gas phase, hence the partial pressure of the Si precursor can be higher without risking the formation of free Si, thus allowing a higher deposition rate. According to the Figure 2b, in the absence of Cl, SiC_(s)_ + Si_(s)_ is deposited (noted by an arrow in Figure 2b) and an increase of (Cl/Si)_(g)_ prohibits the SiC_(s)_ + Si_(s)_ deposition for all temperatures [93]. Nevertheless, for high (Cl/Si)_(g)_, the formation of SiCl_2(g)_ is favored [98], meaning that the concentration of the silicon species Si_(g)_ will decrease, which can lead to the incorporation of carbon species and subsequent formation of C_(s)_ [90].
Figure 2(**a**) SiC composition domains as a function of (C/Si)_(g)_ gas ratio (vertical axis) and of deposition temperature (horizontal axis) for the Si–C–H system, calculated with a thermodynamic model proposed by Chichignoud [88]. Dotted lines correspond to the Si–C–H system with Ar carrier gas, while continuous lines correspond to the same system with H_2_. (**b**) Solid composition as a function of Cl/Si and H/Si partial pressure ratios in the Si–C–H–Cl system for various temperatures according to the thermodynamic model proposed by Chichignoud et al. [93]. Reprinted from “Chlorinated silicon carbide CVD revisited for polycrystalline bulk growth”, G. Chichignoud, M. Ucar-Morais, M. Pons, E. Blanquet, *Surface and Coatings Technology*, Vol. 201, Pages No. 8888–8892, Copyright (2007), with permission from Elsevier.
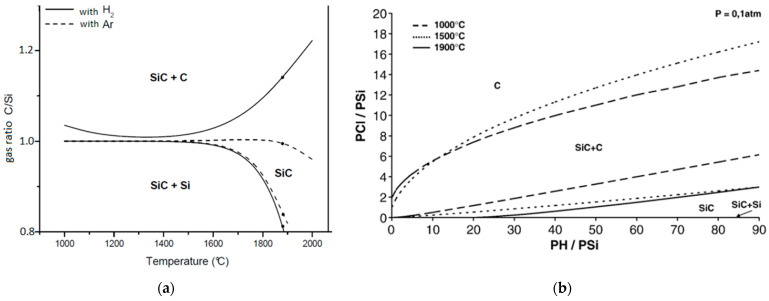


**Pressure effect:** From the point of view of thermodynamics, the pressure alters the efficiency of the gaseous reactions and the amount of the active chemical species into the reactor [98]. The thermodynamic effect of pressure can vary according to the chemistry of the precursors and the chosen growth conditions. Low pressure seems to favor the formation of stoichiometric SiC for the methylsilane/Ar system (C–H–Si system) (Figure 3) [99,100]. Nevertheless, in the case of the (CH_3_SiCl_3_–H_2_)_(g)_ system, increasing the pressure leads to a shift of the optimal temperature for obtaining pure stoichiometric SiC to a higher value [90,91,92].
Figure 3Composition of the deposited solid as a function of T and P observed in the methylsilane/Ar system (re-drawn from [99]). Stoichiometric SiC material is deposited in a limited range of temperature and pressure conditions. High pressure conditions tend to deposit either C-rich for high temperature (>900 °C) or Si-rich for lower temperatures (<900 °C). Below 700 °C there is no deposition of SiC material at any given condition of pressure. (MTLR: mass-transport limited regime; KLR: kinetic limited regime.) The graph has been constructed by experiments and explained by thermodynamic modeling.
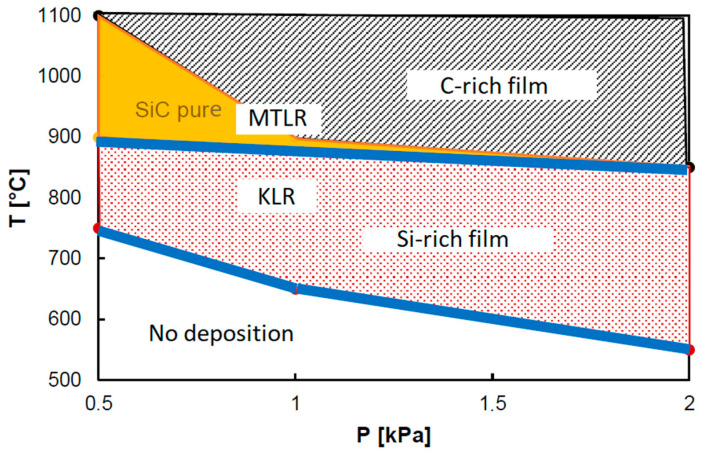


#### 3.1.2. Kinetic Modeling—Gaseous Species

The analysis below is from the work of Danielsson et al. [98], based on quantum chemical modeling and computational fluid dynamics (CFD) calculations, as well as on an analysis of experimental trends. The gas phase kinetic model is proposed mostly for the CVD deposition of epitaxial SiC, but it can be also extended in the low deposition temperatures for polycrystalline SiC. Figure 4 shows the temperature evolution of impingement, equivalently collisional rates of the main gaseous species calculated for both the C–H–Si and the C–Cl–H–Si systems [98].

**Growth Temperature effect:** The temperature influences the formation of the radicals (expected thus to be reactive) for the Si and C deposition on the growing SiC surface [98]. For sufficiently high temperature (>1300 °C) the main impinging C species involved in SiC growth are CH_3(g)_ and C_2_H_2(g)_ (Figure 4a). However, by taking into account the sticking coefficient of these two radical species, CH_3(g)_ is the main species participating in the growth [98]. For Si, the main participating species are Si_(g)_ (>1300 °C), SiH_(g)_ (>1300 °C), and SiCl_(g)_ (>1200 °C) [98]. Despite their high impingement rate, SiCl_2(g)_ and SiHCl_(g)_ do not participate significantly in the growth because of their very low sticking coefficient [98].

**(Cl/Si)_(g)_ effect:** As mentioned above, the purpose for the introduction of chlorine is the reduction of Si co-deposition. In the gas phase, large Si “clusters”, such as Si_2(g)_, Si_3(g)_, Si_5(g)_, and Si_6(g)_, can be seen as embryos for Si solid homogeneous nucleation, and preventing their apparition could lead to less Si co-deposition. From Figure 4, it is obvious that the addition of chlorine to the system does not substantially modify the concentration of carbon gaseous species that contribute to the SiC growth. On the other hand, the amount of Si_2(g)_, Si_3(g)_, Si_5(g)_, and Si_6(g)_ is drastically reduced when chlorine is added, leading to the apparition of SiCl_2(g)_, SiCl_(g)_, and SiHCl_(g)_ (Figure 4b,d) [98]. The SiCl_(g)_ was shown to have a high sticking coefficient on SiC also allowing the deposition of SiC. While present in the highest amount, SiCl_2(g),_ is considered not to be reactive, yet it acts as a silicon “reservoir” in the gas phase that allows the formation of more reactive species (e.g., SiCl_(g)_). Note, however, that chlorine could inhibit the growth of SiC due to the poisoning effect of halogen atoms [101].
Figure 4Impingement rates calculation of the most abundant species in CVD systems, using the calculated partial pressure of the species at thermodynamic equilibrium in the temperature interval 1000−1700 °C. All four graphs correspond to the same two systems; dash-dotted lines correspond to C−H–Si and solid lines to C−Cl−H–Si. (**a**) Hydrocarbons, (**b**) silicon and silicon hydrides, (**c**) Si−C molecules, and (**d**) silicon chlorides [98].
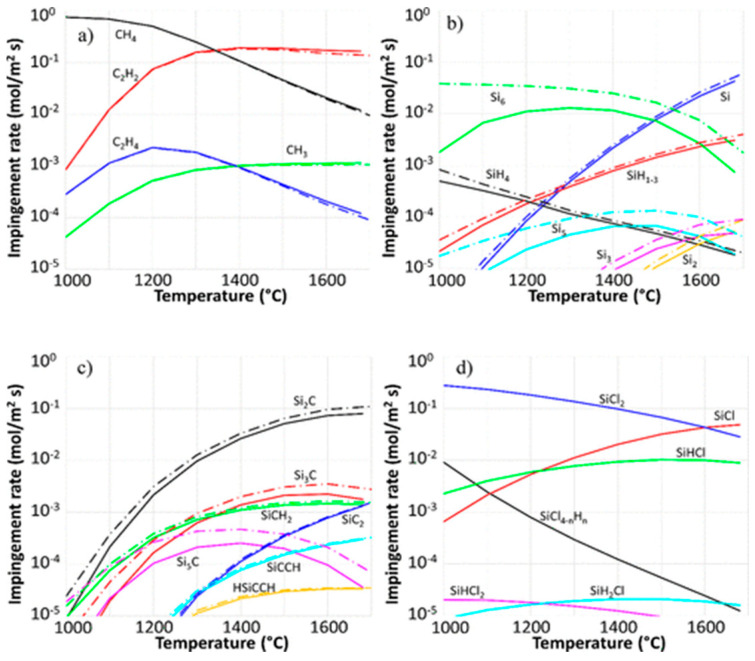


**Pressure effect:** Changing the pressure value results in a variation of the residence time. Higher pressure results in a higher residence time which could possibly vary the mole fractions of the different C and Si species (Figure 5) [98] favoring stable gas molecules at the expense of the more reactive molecules, like CH_3(g)_, that tends to decrease with the increase of residence time. The mole fraction of SiCl_(g)_, the other radical that contributes significantly to the growth of SiC, seems less sensitive to the change of the residence time but could be affected by pressure, due to kinetic effects. Additional work on the kinetic modeling of SiC is needed to further understand the role of pressure on the concentration of gaseous species.
Figure 5Kinetic modeling: species mole fraction vs. residence time for (**a**) (C_2_H_4_ + SiH_4_ + HCl)_(g)_ as precursors, and (**b**) (C_3_H_8_ + SiCl_4_)_(g)_ as precursors. *T* = 1600 °C, total pressure, *p* = 100 mbar, and partial pressure ratios (Si/H_2_)_(g)_ = 0.25, (C/Si)_(g)_ = 1.0, (Cl/Si)_(g)_ = 4 [98].
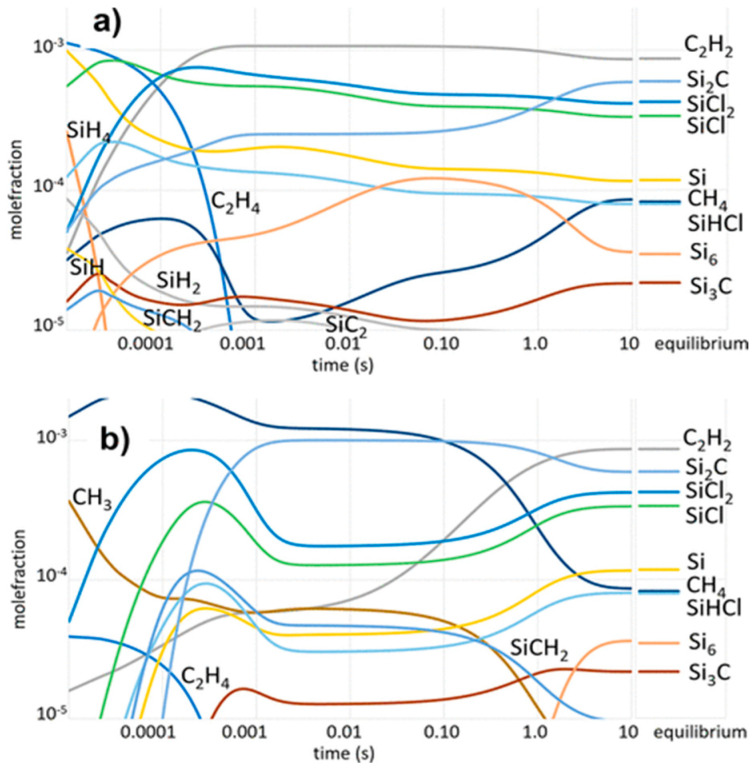


#### 3.1.3. Kinetic Versus Thermodynamic Modeling in the Case of Polycrystalline SiC Growth

Thermodynamics first allows us to explore the feasibility of a deposition process as well as the big trends in terms of deposited solid species (e.g., SiC, C, Si) and the evolution of the gas phase composition. This approach is particularly relevant for high temperature processes where equilibrium of the gas phase can be reached in the time scale of the reactor (residence time), owing to fast kinetics. This is especially true for epitaxial CVD SiC growth in hot-wall reactors where the temperature used is generally high (>1500 °C). However, for moderate temperatures and/or short residence times typical of polycrystalline SiC CVD growth, equilibrium in the gas phase is not necessarily reached in the time scale of the reactor, hence kinetics must be considered, for instance, to evaluate the composition of the gas phase just above the deposition zone which in turn governs the deposition process.

### 3.2. Influence of Growth Conditions on the Growth Rate

**Growth Temperature:**Figure 6 presents the dependence of the growth rate on the temperature for two types of precursors (metalorganic and halide). The growth rate for the C–H–Si system varies from some 0.1 to 10 s of μm/h (Figure 6a), and for the C–Cl–H–Si system it varies between 0.1 µm/h and 100–200 µm/h (Figure 6b). The growth rate increases with temperature following the Arrhenius law [102] in agreement with the kinetic limited regime [103] for all cases except for the studies [81,104,105], where a transition from kinetic limited regime to the mass transport limited regime [103] is observed (Figure 6 and Figure 7). This Arrhenius law results from the fact that both the solid-surface (heterogeneous) and gas-phase (homogeneous) reactions are thermally activated: a higher temperature results in a higher precursor decomposition rate, an increase of the collisional rate, and eventually an increase of the sticking coefficient if the activation energy must be overcome. The observed activation energy is then attributed to one limiting step (single or chain reaction). Based on the experimental results, the activation energies corresponding to the kinetic limited regime (linear part of the curve in Figure 6) can be extracted. The results are reported in Table 2 (for the C–H–Si system) and Table 3 (for the C–Cl–H–Si) system). The activation energy values of the C–H–Si system are lower than that of the C–Cl–H–Si system. Moreover, the range of activation energy in the C–H–Si chemistry is 34–215 kJ/mol, while for the chlorinated chemistry there is a narrower range of activation energy (between 100 and 230 kJ/mol), despite the use of different precursors. The latter narrow activation energy range indicates that there is one limiting step common to all precursor systems which governs the growth rate of SiC in the case of chlorinated chemistry. Additional work coupling kinetic modeling and experiments could help us to more deeply understand this phenomenon.
Figure 6Growth rate vs. reciprocal deposition temperature for (**a**) the C–H–Si system; (**b**) the C–Cl–H–Si system. The references on this graph are reported in Table 2 (for the C–H–Si system) and Table 3 (for the C–Cl–H–Si) system) (References for these graphs: (**a**) Zorman et al. (1997) [73], Orthner et al. (2009) [105], Boo et al. (1995) [106], Cheng et al. (2002) [107], Ohshita (1995) [104], Larkin and Interrante (1992) [108]; (**b**) Gavalas et al. (2024) [10], Chin et al. (1977) [109], Choi et al. (1992) [81], Hu et al. (2019) [110], Kim et al. (1995) [82], So and Chun (1988) [111], Gallou et al. (2023) [8], Liu et al. (2009) [112]). Note that the operating parameters (pressure, flow rates, reactor geometry) were different for each study but kept constant when the deposition temperature was modified.
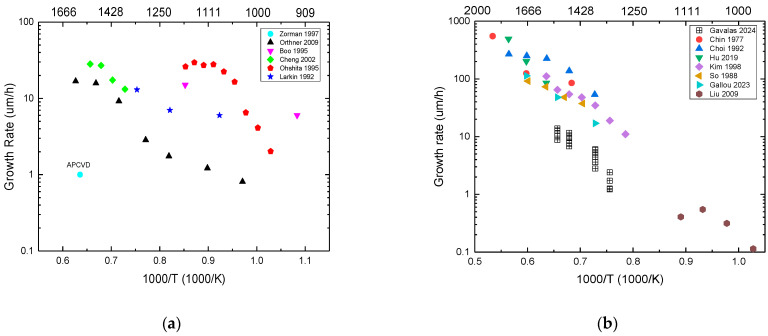
micromachines-16-00276-t002_Table 2Table 2Reported activation energy for the growth of polycrystalline SiC in the C–H–Si system.Reference Activation Energy (kJ/mol)CalculatedTemperature Range (°C)PrecursorsSubstrateZorman et al. [73]No data 1300SiH_4/_C_3_H_8_Polycrystalline SiOrthner et al. [105] 160 750–1325SiH_4/_C_3_H_8_Si (100)Boo et al. [106]33650–900C_2_H_10_Si_2_p-type Si (100)Cheng et al. [107]951100–1250(CH_3_)_3_SiH/NH_3_Si_3_N_4_/Si (111)Ohshita [104]214700–900CH_3_SiH_3_/PH_3_Si (100) Larkin and Interrante [108]36810–1055C_6_H_16_Si_2_Si (100)
micromachines-16-00276-t003_Table 3Table 3Reported activation energy for the growth of polycrystalline SiC in the C–Cl–H–Si system.Reference Activation Energy (kJ/mol)CalculatedTemperature Range (°C)Precursors SubstrateGavalas et al. [10] 1541050–1250SiH_4/_C_3_H_8_/Cl_2_SiO_2_/Si(100)Gallou et al. [8]120900–1400CH_3_SiCl_3_GraphiteHu et al. [110]2301300–1600SiCl_4/_CH_4_GraphiteSo and Chun [111]1901100–1500CH_3_SiCl_3_GraphiteChoi et al. [81]1601100–1500CH_3_SiCl_3_/C_3_H_8_GraphiteKim et al. [82]1171050–1300CH_3_SiCl_3_GraphiteChin et al. [109]1001150–1600CH_3_SiCl_3_No dataLiu et al. [112]137700–850 SiH_3_CH_3_/SiH_2_Cl_2_Si (100)Desenfant et al. [100] *150800–1150C_2_H_3_Cl_3_SiGraphite* Not included in Figure 6b.

**(C/Si)_(g)_:** The (C/Si)_(g)_ inlet ratio seems to influence the growth rate but the reported results are contradictory. For instance, in [81] when (C/Si)_(g)_ increases, the growth rate increases, as seen in Figure 7a. The same observation was made in [95,113]. Conversely, in [114] when (C/Si)_(g)_ increases, the growth rate decreases, as shown in Figure 7b. Note that the difference in the latter study is the use of Si–C–Cl–H chemistry while the aforementioned references use a C–H–Si chemistry. So, this discrepancy can probably be related with the concentration of the main carbon and silicon reactive species in the gaseous phase just above the deposition zone, which is linked with the precursors’ flow rate and nature. Excess Si_(g)_ (when local (C/Si)_(g)_ < 1) or C_(g)_ (when local (C/Si)_(g)_ > 1) reactive species can modify the kinetics of different homogeneous and surface reactions, changing the growth rate accordingly.
Figure 7(**a**) (Left) Deposition rate of polycrystalline SiC thin films vs. reciprocal temperature at different (C/Si)_(g)_ ratio values [81]. Reprinted from “The Effects of C3H8 on the chemical vapor deposition of silicon carbide in the CH_3_SiCl_3_ + H_2_ System”, B.J. Choi, S.H. Jeun, D.R. Kim, *Journal of the European Ceramic Society*, Vol. 9, Pages No. 357–363, Copyright (1992), with permission from Elsevier. (**b**) Growth rate of polycrystalline SiC vs. the (Si/C)_(g)_ inlet ratio at deposition temperature 1373K [114]. Reprinted from “Model of morphology evolution in the growth of polycrystalline β-SiC films”, J. Yun, D.S. Dandy, *Diamond and Related Materials*, Vol. 9, Pages No. 439–445, Copyright (2000), with permission from Elsevier.
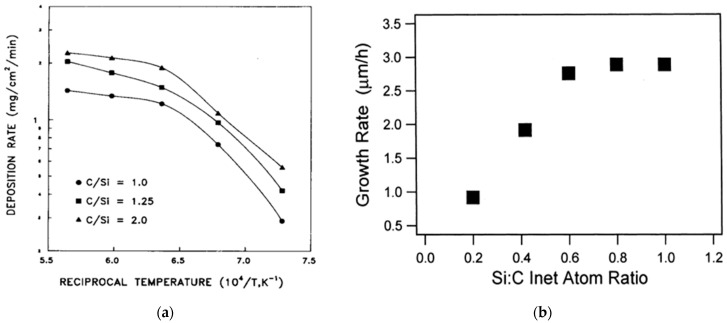


**(Cl/Si)_g_**: A higher (Cl/Si)_(g)_ results in a lower growth rate (Figure 8) [74,96,115]. This can be related to the increase of the SiCl_2(g)_ concentration at the expense of more active radical species like SiCl_(g)_. In addition, chlorine can inhibit SiC growth by a poisoning effect [74,96,98,101]. Additional kinetic modeling studies are needed, notably to clarify the effect of SiCl_2(g)_.
Figure 8Growth rate vs. (Cl/Si)_(g)_ (References for this graph: Papasouliotis and Sotirchos (1999) [72], Gao et al. (1998) [96], and Ohshita (1989) [115]). Operating parameters (pressure, flow rates, temperature, reactor geometry) were different for each study but kept constant when the (Cl/Si)_(g)_ was modified.
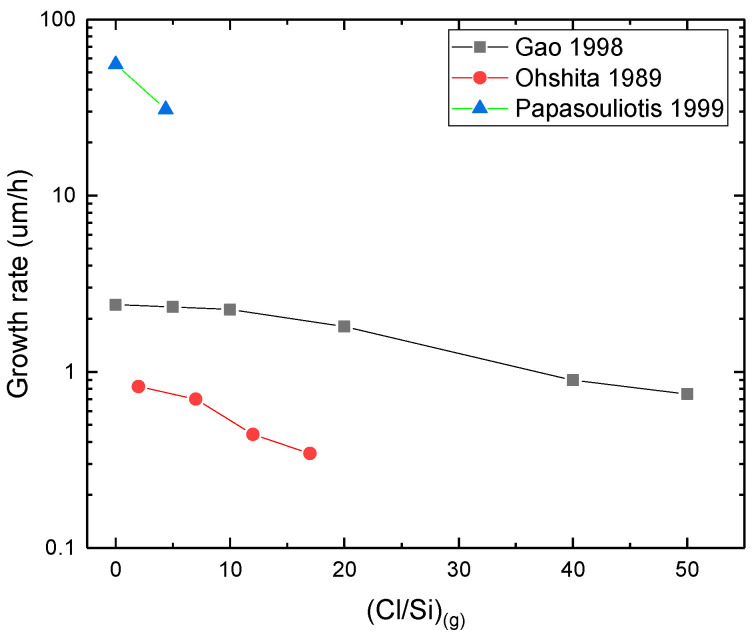


**Deposition chamber pressure:** The increase of pressure can increase [99,100,104,111,116] or slightly decrease the growth rate [26,105,112,117]) depending on the chemical system, the residence time, and the operation regime of the reactor (kinetic or mass-limited). Note that the variation with pressure in [26,104,105,112,116,117] has been studied for one constant temperature. The main effect observed from pressure increase and the corresponding interpretation are presented below. (1)The pressure increase moves the transition point from the kinetic limited regime to the mass transport limited regime at lower temperatures [99,100,111] (see Figure 9a,b). This is mainly because increasing pressure results in (a) an increasing adsorption rate due to higher collisional frequency and/or (b) a decrease of the diffusion coefficient.(2)According to [104,111,116], a strong increase of the growth rate with pressure is observed in the case of kinetic limited regime (Figure 9c) due to an enhancement of the intermolecular reactions in the gas phase and of the adsorption rate (induced by a higher collision frequency). However, a constant or a slightly decreased growth rate with pressure was found in [26,112].(3)A minor decrease of the growth rate with pressure has been observed in the case of the mass transport limited regime with some variation on the effect amplitude according to the employed chemical system [99,100,105,117] (see Figure 9). This behavior can be explained by a decrease of the mole fraction of the radical species [105,117] which might partially compensate the increase of the collisional rate with pressure.
Figure 9Effect of the deposition pressure in the evolution of the growth rate of SiC according to the deposition temperature: (**a**) for the methlysilane (MS) (Si–C–H chemical system); (**b**) for the VTS (Si–C–Cl–H chemical system) [99]. Pressure increases the growth rate of SiC in the kinetic limited regime, while for the mass transport limited regime the evolution of growth rate is less sensitive to a pressure change and depends on the chemical system used. (**c**) Pressure dependence of the polycrystalline SiC growth rate as a function of the growth regime (References for this graph: Figueras et al. (1991) [117], Orthner et al. (2009) [105], Boo et al. (2003) [116], Liu et al. (2009) [112], Fu et al. (2014) [26], Ohshita (1995) [104], So and Chun (1988) [111]). Note that the operating parameters (flow rates, temperature, reactor geometry) were different for each study but kept constant when the pressure was modified.
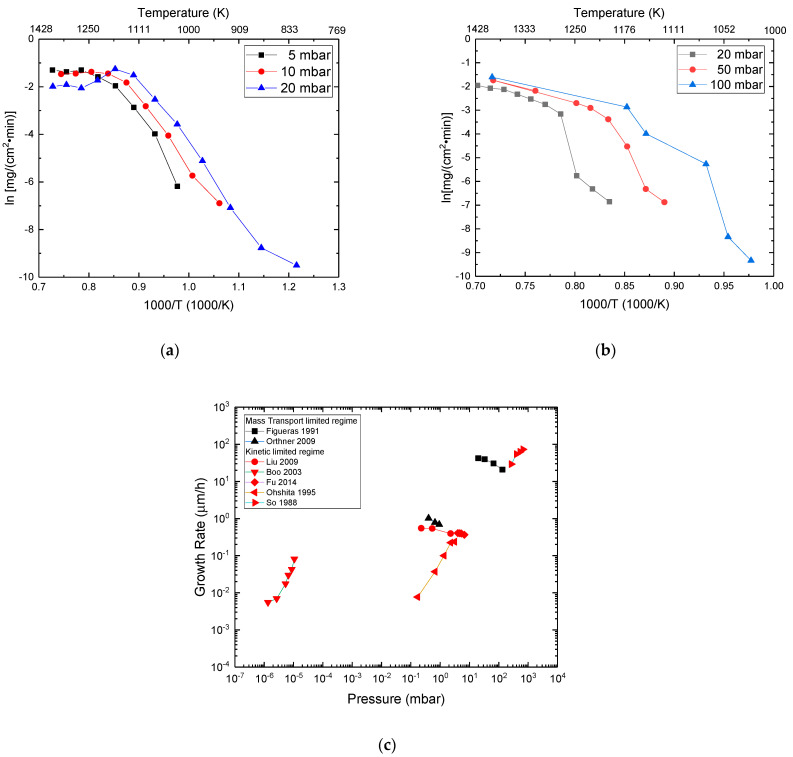
**Doping**: The effect of doping atoms on the growth rate is not fully clear, as contradictory results have been reported. Indeed, in [118,119] the increase of ammonia or nitrogen flow rate decreases the deposition rate. By contrast, other studies show the opposite results [19,26], possibly due to the inclusion of amorphous phases. A third study showed that the N_2(g)_ flow rate does not affect the deposition rate [120]. The growth rate changes with the doping variation, which has been described as a possible result of the three following effects. Firstly, a possible co-deposition of Si_3_N_4_ affecting the overall growth rate [26]. This can be true especially for low growth temperatures (<900–1000 °C) where Si_3_N_4_ can be stable together with SiC, under otherwise typical N-doped SiC growth conditions [121,122]. However, it is not clear if it increases the deposition rate [26] or decrease it [118]. Secondly, a change of the Fermi energy level induced by varying doping concentration can enhance the surface stability due to the negatively charge transferred electron donors from the gas phase to the film’s surface, lowering thus, the surface reactivity and the corresponding growth rate [123]. Thirdly, an increase of surface coverage by impurities (in this case nitrogen atoms) inhibiting the SiC formation [124].

### 3.3. Doping of Polycrystalline SiC LPCVD

**N-doping**: Nitrogen is the most usual donor atom for the n-doping of polycrystalline SiC films by LPCVD growth. Precursors, such as N_2(g)_ and NH_3(g)_, are used as sources of nitrogen. It is shown (Figure 10) that, for the same amount of nitrogen incorporated in SiC, a lower (N/C)_g_ ratio is required when using NH_3_ compared to N_2_ In other words, for a constant precursor flow rate, a lower NH_3_ flow rate is required to achieve the same nitrogen content in polycrystalline SiC as with N_2_. This is related to the higher reactivity of NH_3_, essentially due to the lower bond energy of N–H_(g)_ compared to N–N_(g)_.
Figure 10Nitrogen concentration of polycrystalline SiC as a function of (N/C)_(g)_ (References for this graph: Wijesundara et al. (2002) [118], Zhang et al. (2007) [125], Lei et al. (2009) [126], Lai et al. (2020) [119], Latha et al. (2015) [127], Tabata et al. (2010) [128].) Note that the operating parameters (pressure, flow rates, temperature, reactor geometry) were different for each study but kept constant when the (N/C)_g_ ratio was modified.
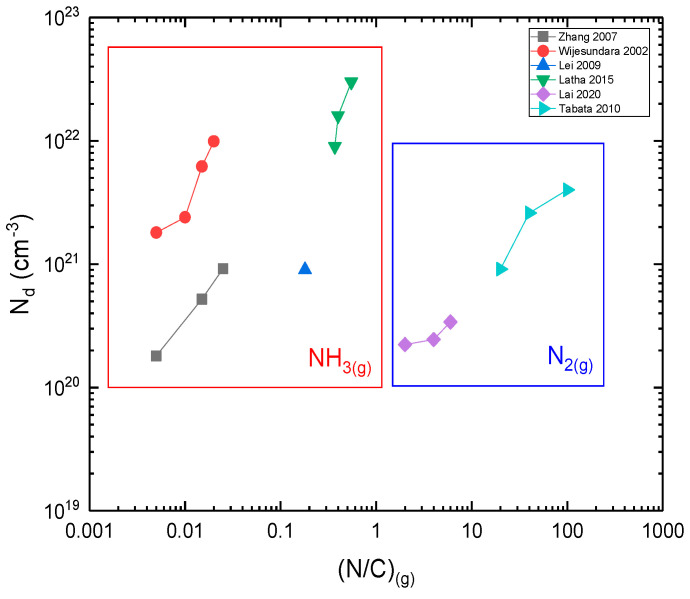
**P-doping:** LPCVD p-type polycrystalline SiC films are deposited similarly (in terms of deposition temperature and pressure) with the n-type SiC. The acceptor atom is usually aluminum (Al) or boron (B). Precursors such as Al(CH_3_)_3(g)_ or B_2_H_6(g)_ are used for the p-doping of polycrystalline SiC [20,83,84,125,129].**Doping by site competition:** Site competition has been described as an effective SiC doping control method, mainly employed in thermal CVD, that relies on properly adjusting the (C/Si)_(g)_ ratio into the reactor to control the amount of incorporated dopant atoms (N for n-doping or Al for p-doping) into the SiC lattice [130]. This technique is based on the principle of site competition between the N and C for the C sites and between Al and Si for the Si sites [130]. Therefore, the (N/C)_g_ and (Al/Si)_g_ ratios are also of key importance. Even though this method has been extensively studied for monocrystalline SiC, to our knowledge there is not yet a systematic study for the polycrystalline SiC films. Nevertheless, the site competition between N and C, has been proposed by Sun et al. [79], Wijesundara et al. [118], Latha et al. [127], and Tabata et al. [128], as an explanation of their ex situ XPS results of nitrogen-doped polycrystalline SiC. Indeed, their results indicate that the incorporated atomic N increases with NH_3(g)_ flow rate and is inversely proportional with the atomic C, into the SiC lattice (Figure 11), indicating that N substitutes C and bonds with Si, even at a very high concentration.
Figure 11XPS measured elemental percentage of Si and C vs. the amount of ammonia inlet during n-doping of polycrystalline SiC [118]. Reprinted from “Nitrogen doping of polycrystalline 3C–SiC films grown by single-source chemical vapor deposition”, M.B.J. Wijesundara, C.R. Stoldt, C. Carraro, R.T. Howe, R. Maboudian, *Thin Solid Films*, Vol. 419, Pages No. 69–75, Copyright (2002), with permission from Elsevier.
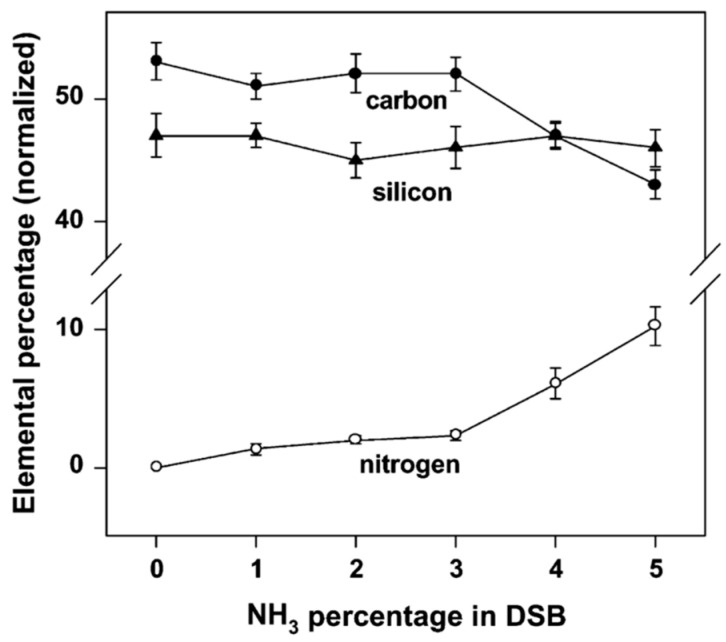


### 3.4. Summary Tables for Experimental CVD Growth Conditions

The following tables summarize the experimental growth conditions and the corresponding measurement results of polycrystalline SiC properties for the C–H–Si system (Table 4) and the C–Cl–H–Si system (Table 5).

## 4. Relation Between the Growth Conditions and the Materials Properties—Experimental Work

### 4.1. Structural Characterization

#### 4.1.1. Composition of the Deposited Film

A major effort [96,115] has been dedicated to the elimination of Si co-deposition by varying the growth conditions (mainly introducing Cl-based chemistry [81,95,109,112,115]). As a result, various studies using elemental analysis techniques, like XPS [134,135], EPMA [95,109,132,134,147] and Auger spectroscopy [96,108,115], showed cases of Si-rich SiC films. Fewer studies have focused on an eventual problem of C-agglomerations eventually due to the widely accepted idea that the excess of C in the gas phase would not influence the deposited film, as C reacts with the traces of oxygen in the growth chamber retaining it in the gas phase. However, polycrystalline SiC films with an excess of C have also been observed in several cases [97,99,100,135]. Also note the fact that Si excess is presented mostly in the form of crystallites resulting in an easier confirmation (e.g., by XRD) of the presence of Si agglomerations. The effect of growth parameters on the stoichiometry of deposited polycrystalline SiC is discussed below.

**Growth Temperature:** Experimental work on the C–Si–H chemical system [78,104,105,106,108,132,134,135] shows that stoichiometric polycrystalline SiC_(s)_ was obtained in most cases (Figure 12a). The authors of [108] observed that an increase of deposition temperature reduces the Si_(s)_ content of the deposited film, leaving only a stoichiometric SiC_(s)_ deposit. Another study showed that an increase of the deposition temperature ends up in C-rich films [135]. On the contrary, for two studies [132,134], the increase of the deposition temperature lowers the C_(s)_ content in the deposited film, ending with a stoichiometric SiC_(s)_ material.In the chlorinated C–Cl–Si–H chemical system [81,95,109,115] it is shown that the increase of temperature reduces the deposition of Si_(s)_ clusters (Figure 12b and Figure 13a). The temperature range for removing the Si clusters is between 1100 °C and 1400 °C, depending on studies and growth conditions [95,100,109]. According to the same studies, the C_(s)_ content of the film is increased with the deposition temperature, in agreement with the thermodynamics studies [88,91,92,94].
Figure 12Chemical composition ratio (C/Si)_(s)_ evolution with deposition temperature for (**a**) the C–H–Si chemical system (References for this graph: Orthner et al. (2009) [105], Boo et al. (1995) [106], Ohshita (1995) [104], Clavaguera-Mora et al. (1997) [132], Larkin and Interrante (1992) [108], Wijesundara et al. (2003) [78], Chiu and Hsu (1994) [134], Liu et al. (2022) [135]); (**b**) for the C–Cl–H–Si system (References for this graph: Chin et al. (1977) [109], Kim et al. (1995) [82], Trevino et al. (2005) [145], Roper et al. (2009) [148], Kuo et al. (1990) [95], Fu et al. (2014) [26], Fu et al. (2005) [152]). Note that the operating parameters (pressure, flow rates, reactor geometry) were different for each study but kept constant when the deposition temperature was modified. The aim of these compilations was to detect any eventual systematic effect of temperature on the (C/Si)_(s)_ ratio.
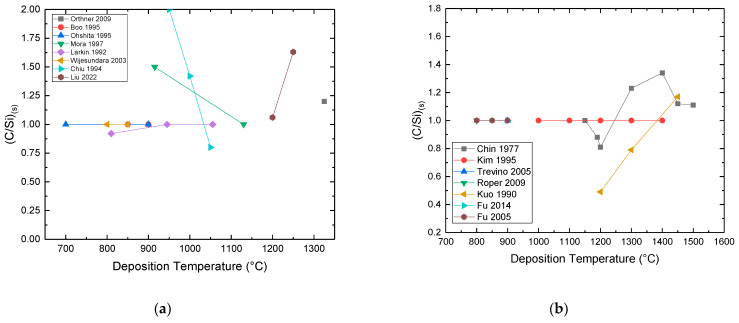


**(C/Si)_(g)_:** While the (C/Si)_(g)_ ratio has been extensively studied in the bulk and epitaxial growth of SiC, due to its importance as a process parameter to control the polytype stability, the growth rate, the growth front stability, and also the doping [6,153], there are a few similar studies for polycrystalline SiC, mainly in the case of the C–Cl–H–Si system [81,96,112,115], while one reference can be found in the C–H–Si system [135]. Experiments agree qualitatively with the thermodynamics prediction and co-deposition of Si_(s)_ and C_(s)_, which is observed for low and high (C/Si)_(g)_ value, respectively. In Ref. [135], the authors demonstrated that the increase of the inlet (C/Si)_(g)_ results in a mixed deposit of SiC_(s)_ and free C_(s)_. Liu et al. [112], Gao et al. [96], Ohshita [115], and Choi et al. [81] show that (C/Si)_(g)_ ≤ 1 at the inlet can lead to a Si_(s)_ + SiC_(s)_ deposition (Figure 13). Also, in the methyltrichlorosilane + H_2(g)_ system, a too low temperature can lead to the co-deposition of Si film [81,95,96,97]. An extra source of carbon (therefore increasing (C/Si)_(g)_) can lead to the deposition of pure SiC. However, a careful tuning of the extra carbon source in the gas phase is necessary to avoid the co-deposition of C_(s)_ in the SiC_(s)_ film [81]. Finally, note that, for the epitaxial growth of SiC, Leone et al. [154] have shown, by gas phase modeling for many different chemical systems, that (C/Si)_(g)_ can be as low as 0.1–0.3, close to the substrate for stoichiometric growth, while the corresponding inlet ratio could be much higher. Obviously, the knowledge of the local values is of paramount importance when discussing the effect of gas composition on SiC growth from different studies. Such studies are today missing for the polycrystalline growth of SiC.
Figure 13(**a**,**b**) XRD diffractograms of (CH_3_SiCl_3_ + C_3_H_8_/Ar)_(g)_ system for (C/Si)_(g)_ = 2 and 1, respectively. In (**a**) Si peaks are not observed whatever the temperature, indicating that the deposited film is either an SiC_(s)_ film or SiC_(s)_ with C_(s)_ [81] as the high temperature renders improbable the deposition of a-Si. In (**b**) Si peaks are reduced along with the deposition temperature, allowing the growth of stoichiometric SiC [81]. Reprinted from “The Effects of C_3_H_8_ on the chemical vapor deposition of silicon carbide in the CH_3_SiCl_3_ + H_2_ System”, B.J. Choi, S.H. Jeun, D.R. Kim, *Journal of the European Ceramic Society*, Vol. 9, Pages No. 357–363, Copyright (1992), with permission from Elsevier.
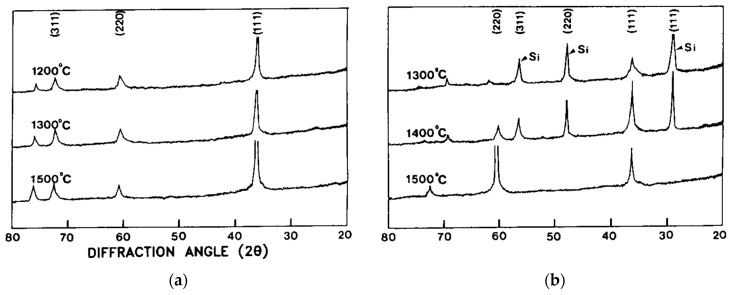


**(Cl/Si)_(g)_:** It has been shown that the increase of the (Cl/Si)_(g)_ ratio prevents the deposition of Si clusters in Si-rich ((C/Si)_(g)_ < 1) gaseous conditions [115]. Another report shows that the increase of the (Cl/Si)_(g)_ ratio results in more stoichiometric SiC film, either in C-rich or Si-rich gaseous conditions (Figure 14) [96]. Note also that the authors of [96] observed a reduction of C-excess when Cl is included in the precursors. In addition to the above described mechanism (see thermodynamics/kinetics section above) of Si_(g)_ retention in the gas phase by the Cl and/or limitation of the Si_(g)_ gas phase nucleation [98], a high Cl concentration can also etch away the extra deposited Si [96,115].
Figure 14(C/Si)_(s)_ solid composition ratio dependence on the (Cl/Si)_(g)_ (References for this graph: Gao et al. (1998) [96], Ohshita (1989) [115]). In both cases, the (C/Si)_(s)_ ratio was measured by Auger electron spectroscopy (AES). Note that the operating parameters (pressure, flow rates, temperature, reactor geometry) were different for each study but kept constant when the (Cl/Si) ratio was modified.
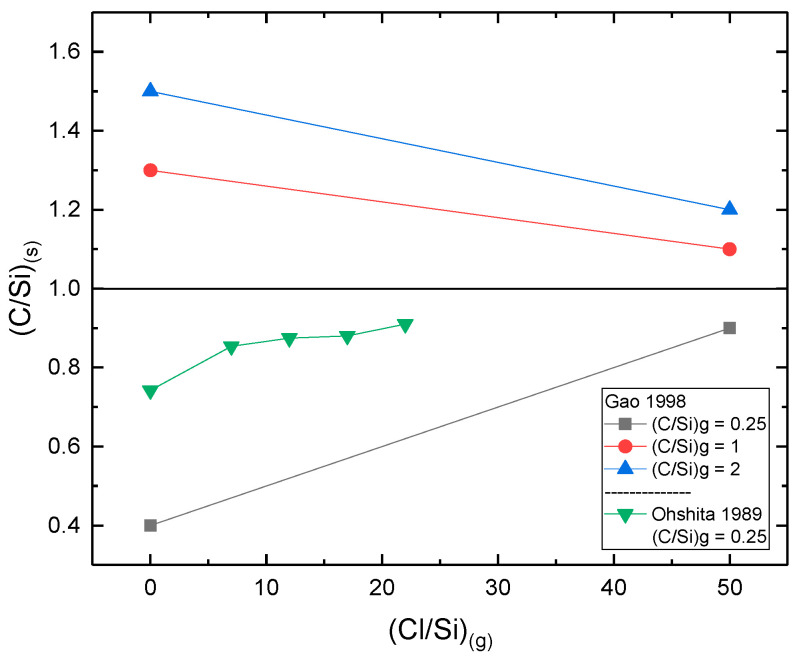


**Deposition chamber pressure:** As discussed previously in Section 2.2, the variation of pressure can have multiple effects (on reaction kinetics, gas flow, residence time, diffusion coefficients) which make the understanding of pressure on composition of the deposited film very complicated. Experimentally, there are very limited studies on the effect of pressure on the composition of deposited polycrystalline SiC film. One experimental study in the range of 18 mbar to 700 mbar reports that high pressures result in Si-rich films, while lower pressures result to C-rich films for the (H_2_/CH_3_SiCl_3_)_(g)_ system (Figure 15) [109]. Stoichiometric SiC films can be deposited for intermediate values of pressure. The stability domain for stoichiometric SiC is also dependent on the growth temperature and the dilution ratio (H_2_/CH_3_SiCl_3_)_(g)_. Once again, these experimental results are not easy to comment on without knowledge of the local growth condition, i.e., just above the substrate.
Figure 15Chemical composition of polycrystalline SiC thin films regarding the substrate temperature, deposition pressure and H_2_/CH_3_SiCl_3_ fraction [109]. Black region shows the growth conditions for the deposition of stoichiometric SiC films. Reprinted from “The structure of chemical vapor deposition silicon carbide”, J. Chin, P.K. Gantzel, and R.G. Hudson, *Thin Solid Films*, Vol. 40, Pages No. 57–72, Copyright (1977), with permission from Elsevier.
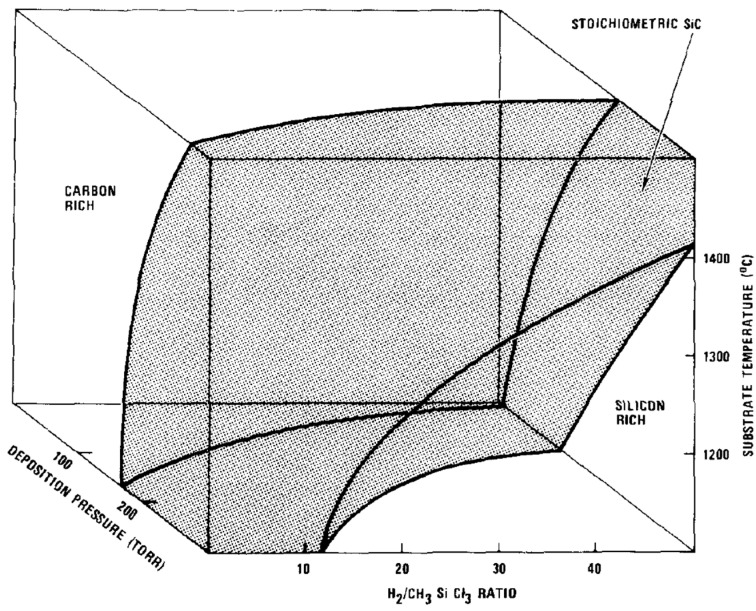


#### 4.1.2. Crystallite, Grain Size, and Surface Roughness

The grain/crystallite size have been investigated by surface SEM [8,10,81,96,111,112,116,117,131,133,144,147], cross-section SEM [8,133], AFM [26,107,112,131,133,147], and XRD [8,10,26,81,96,112,116,131,147]. In the following discussion, the term crystallite is used for the extracted size from XRD, while in all other cases (SEM, AFM) the term grain is employed. As many papers report plane view SEM images, and not from the cross-section of the film, we tried to use them for evaluating the dependence of the grain size as well as surface roughness from the deposition conditions. However, an inherent difficulty with this procedure is that the reported images correspond to different thickness or an equivalent coalescence stage. In addition, this method gives a rough estimation of the average grain size diameter value, ignoring the information of the grain size in the bulk region of the film, which could be given by a cross-section observation. The XRD crystallite size using the standard Bragg–Brentano configuration refers to the average size of the coherent domain (“perfect crystal”) in the direction perpendicular to the surface of the deposit. Owing to these considerations, comparing crystallite size and grain size can lead to misinterpretation.

According to the CVD growth theory, the grain formation follows the heterogeneous nucleation theory on top of a solid surface [102], and the following analysis of the reported experimental results will consider these theoretical expectations.

**Thickness:** It has been shown that the average lateral grain size is increased with the thickness of the film [105,114,126,133], up to certain value, and then tends to a saturation point [114,133] (see Figure 16). Indeed, at the beginning of the growth, the grain size increases proportionally with the film’s thickness; however at a certain point, the crystallite size remains nearly constant [114,133]. It is shown that the RMS surface roughness increases with the film’s thickness (Figure 17), which is related with the increase of the average grain size as the deposition proceeds [105,114,126,133].
Figure 16Evolution of columnar grain size with the polycrystalline SiC film thickness from (**a**) [114] Red circle indicates the exponential factor value calculated from the model of [114], which predicts the grain size evolution with the film’s thickness. Reprint from “Formation of <111> fiber texture in β-SiC films deposited on Si(100) substrates”, V. Radmilovic, U. Dahmen, D. Gao, C.R. Stoldt, C. Carraro, R. Maboudian, *Diamond & Related Materials*, Vol. 16, Pages No. 74–80, Copyright (2007), with permission from Elsevier. (**b**) [133]. Reprinted from “Model of morphology evolution in the growth of polycrystalline β-SiC films”, J. Yun, D.S. Dandy, *Diamond and Related Materials*, Vol. 9, Pages No. 439–445, Copyright (2000), with permission from Elsevier.
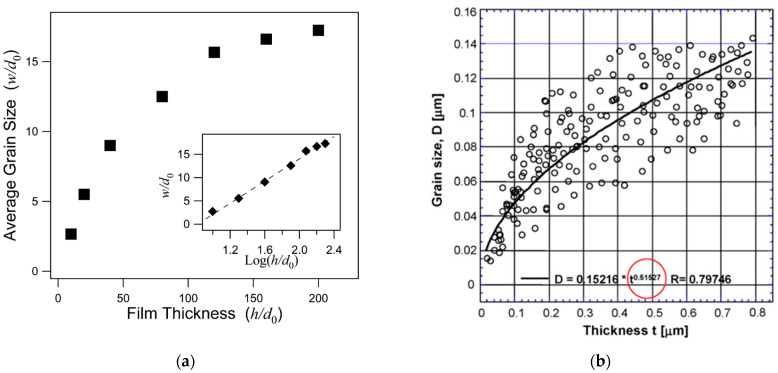

Figure 17AFM measured RMS surface roughness of polycrystalline 3C-SiC films grown on Si (100) at 850 °C as a function of film’s thickness (data taken from [133]).
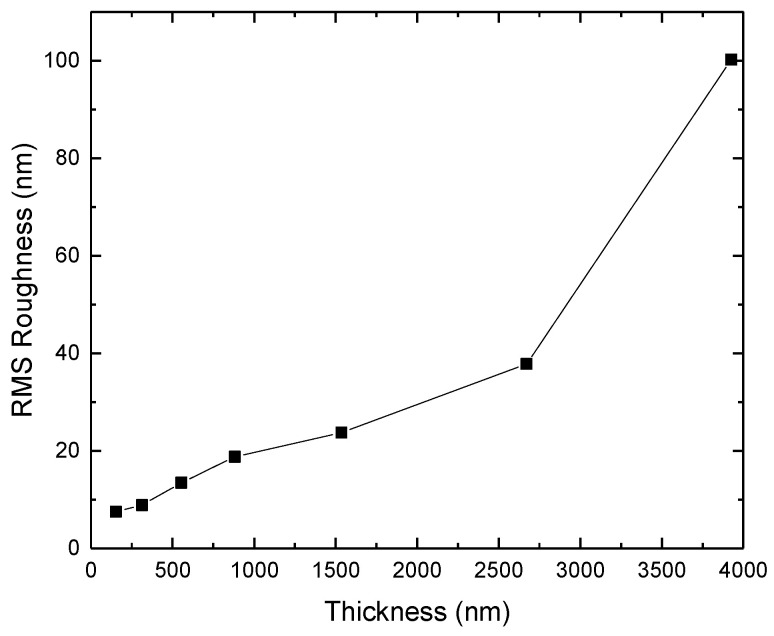


**Growth Temperature**: An increase of temperature provides larger and more oriented grains (Figure 18), since higher deposition temperatures decrease the saturation density of the nuclei according to the nucleation theory. The lower density of the nuclei will allow for more space for islands to grow. As a result, the grain size (width) will be greater independent of the chemistry (C–H–Si (131) and C–Cl–H–Si (111)) and the substrate used (Si (100), SiO_2_, graphite, Al_2_O_3_) [8,10,26,81,111,112,116,131]. In addition, increasing the temperature could increase the mobility of the adsorbed species resulting in higher surface diffusion distance. The desorption of adatoms must also be considered for a higher (>1500 °C) temperature where the single crystalline growth region starts [155]. Many studies [10,107,131,132,152] have reported that the surface roughness increases with the growth temperature (Figure 19). Higher temperatures allow the growth of bigger grains/crystallites resulting in rougher surfaces, in terms of RMS value.
Figure 18(**a**) SEM backscattered electron cross sectional images of polycrystalline SiC grown at (3) ~1100 °C, (6) ~1230 °C, (9) ~1380 °C, (10.5) ~1400 °C, (12) ~1460 °C, (15) ~1500 °C [8]. Blue lines indicate an abrupt junction between two groups of grains. The red circles shows the formation of micro-twins and stacking faults in the [111] direction. Reprinted from “Evidence of twin mediated growth in the CVD of polycrystalline silicon carbide”, Y. Gallou, M. Dubois, A. Potier, D. Chaussende, *Acta Materialia*, Vol. 259, Pages No. 119274, Copyright (2023), with permission from Elsevier. (**b**) SEM images of surface morphology of nitrogen doped polycrystalline SiC grown at (a) 1050 °C, (b) 1100 °C, (c) 1200 °C, (d) 1250 °C [10].
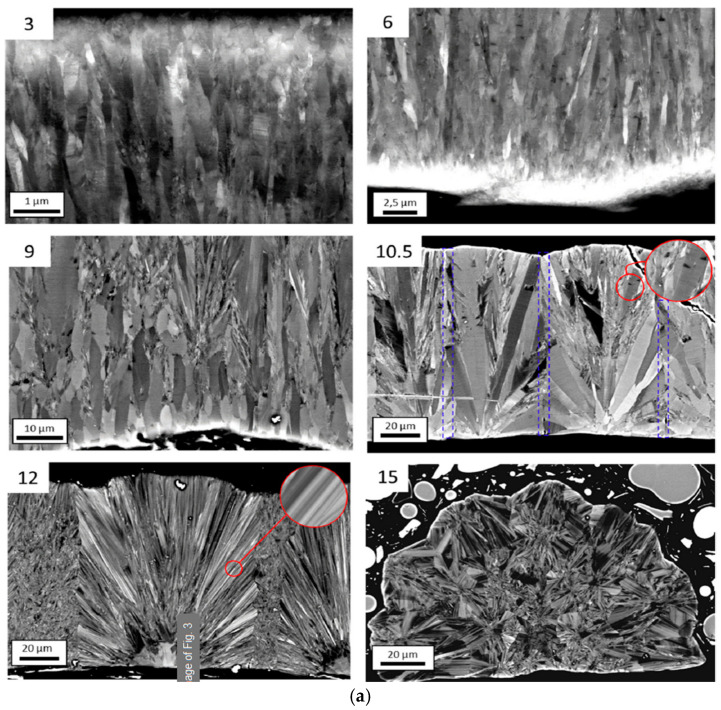

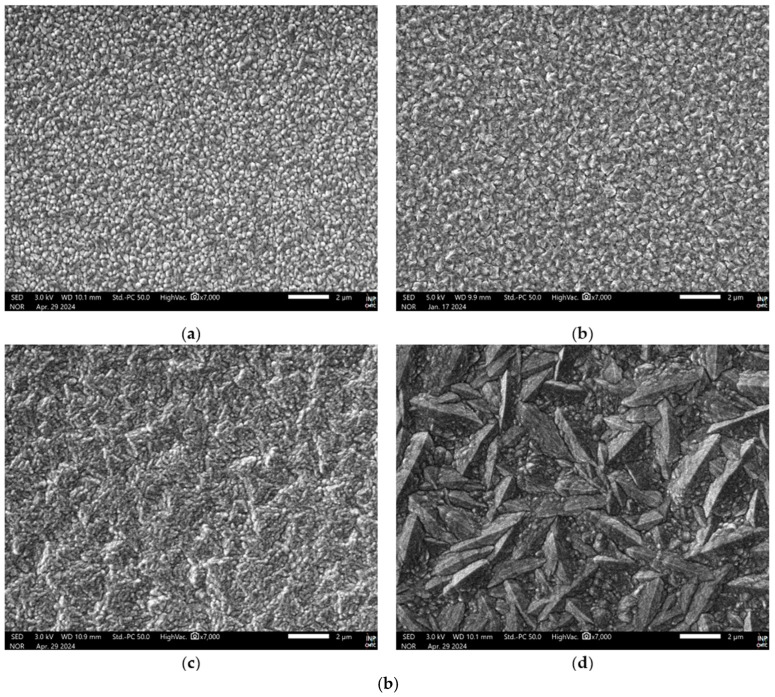
Figure 19RMS surface roughness of polycrystalline SiC evolution with deposition temperature. (References for this graph: Gavalas et al. (2025) [10], Cheng et al. (2002) [107], Fu et al. (2005) [152], Stoldt et al. (2002) [131], Clavaguera-Mora et al. (1997) [132]). The corresponding film thickness has not been reported in most cases. Note that the operating parameters (pressure, flow rates, reactor geometry) were different for each study but kept constant when the deposition temperature was modified.
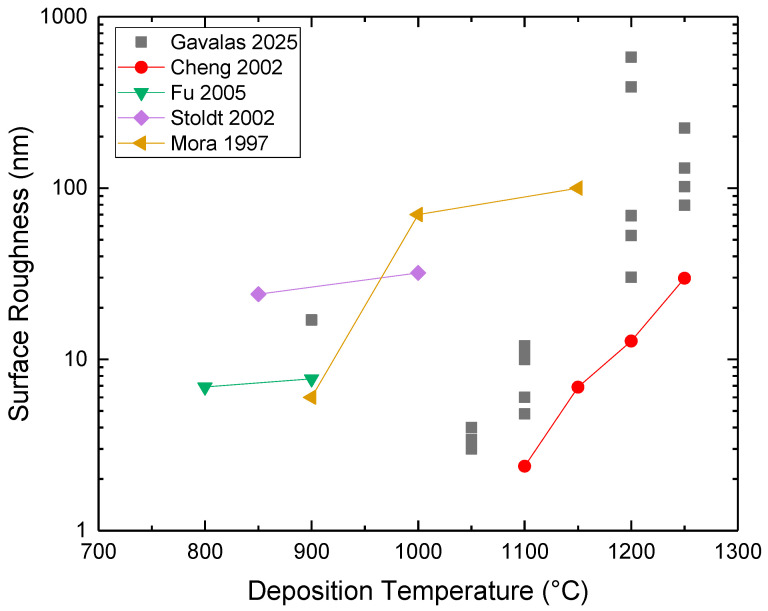


**(C/Si)_(g)_**: It is not easy to interpret the bibliographical results on the effect of the (C/Si)_(g)_ ratio on the grain size and the surface morphology. For instance, according to [81], a high enough increase (C/Si)_(g)_ > 1) of the (C/Si)_(g)_ ratio at the inlet results in an increase of grain size (Figure 20a) due to the suppression of Si co-deposition. Indeed, the co-deposition of Si_(s)_ or C_(s)_ in the SiC layer is believed to enhance the nucleation density of SiC crystals, or equivalently limiting the grain size [81,97].
Figure 20(**a**) SEM micrographs for (a) (C/Si)_(g)_ = 1.0, (b) (C/Si)_(g)_ = 1.25, (c) (C/Si)_(g)_ = 2.0, and (d) (C/Si)_(g)_ = 2.5. The total flow rate = 1600 sccm, MTS mole ratio = 0.01, deposition temperature = 1300 C, and graphite was used as substrate [81]. Reprinted from “The Effects of C_3_H_8_ on the chemical vapor deposition of silicon carbide in the CH_3_SiCl_3_ + H_2_ System”, B.J. Choi, S.H. Jeun, D.R. Kim, *Journal of the European Ceramic Society*, Vol. 9, Pages No. 357–363, Copyright (1992), with permission from Elsevier. (**b**) Plane view SEM images of SiC films deposited with (a) no DCS, (b) 0.10 DCS fraction, (c) 0.14 DCS fraction, (d) 0.16 DCS fraction, (e) 0.18 DCS fraction, (f) 0.31 DCS fraction, and (g) 0.47 DCS fraction [147]. Note that the DCS contains only Si, hence higher DCS fraction decreases the (C/Si)_(g)_ ratio at inlet. Reprinted from “Characterization of polycrystalline 3C-SiC films deposited from the precursors 1,3-disilabutane and dichlorosilane”, C.S. Roper, V. Radmilovic, R.T. Howe, R. Maboudian, *Journal of Applied Physics*, Vol. 103, Page N° 084907, 2008 with the permission of AIP Publishing.
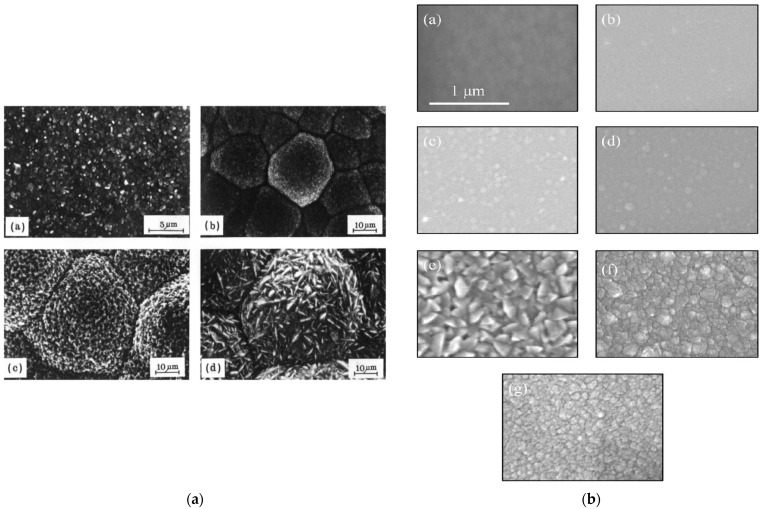


In Ref. [147], the authors showed a more complicated behavior as they varied both the (C/Si)_(g_) and the (Cl/Si)_(g)_ ratios by adding DCS to DSB. It seems that for low DCS flow rates the (C/Si)_(g_ ratio controls the surface roughness and the associated grain size as indicated by their increase. However, at even higher DCS flow rates the (Cl/Si)_(g)_ ratio takes over (see below) with a decrease of the surface roughness and the associated grain size.

**(Cl/Si)_g_:** It has been shown by the authors of [96] that an increase of the (Cl/Si)_(g)_ ratio decreases the surface roughness and improves the crystallinity. Furthermore, plane-view SEM observations show smaller-in-diameter surface grains and lower roughness with increasing (Cl/Si)_(g)_ [96] (see Figure 21), [115] (use of HCl_(g)_), [144] (use of DCS_(g)_), while a complicated behavior has been reported by [147] (use of DCS_(g)—_see the above section). Note that a mirror-like polycrystalline SiC surface has been obtained at high HCl flow rates [115].
Figure 21SEM images of the surfaces of SiC thin films for various (Cl/Si)_(g)_ ratios: (**A**) (Cl/Si)_(g)_: 0 (**B**) (Cl/Si)_(g)_: 5 (**C**) (Cl/Si)_(g)_: 40 (**D**) (Cl/Si)_(g)_: 50 [96]. No significant change is observed between (**C**) and (**D**) cases. Reprinted from “Low-temperature chemical-vapor deposition of 3C-SiC films on Si (100) using SiH_4_–C_2_H_4_–HCl–H_2_”, Y. Gao, J.H. Edgar, J. Chaudhuri, S.N. Cheema, M.V. Sidorov, D.N. Braski, *Journal of Crystal Growth*, Vol. 191, Pages No. 439–445, Copyright (1998), with permission from Elsevier.
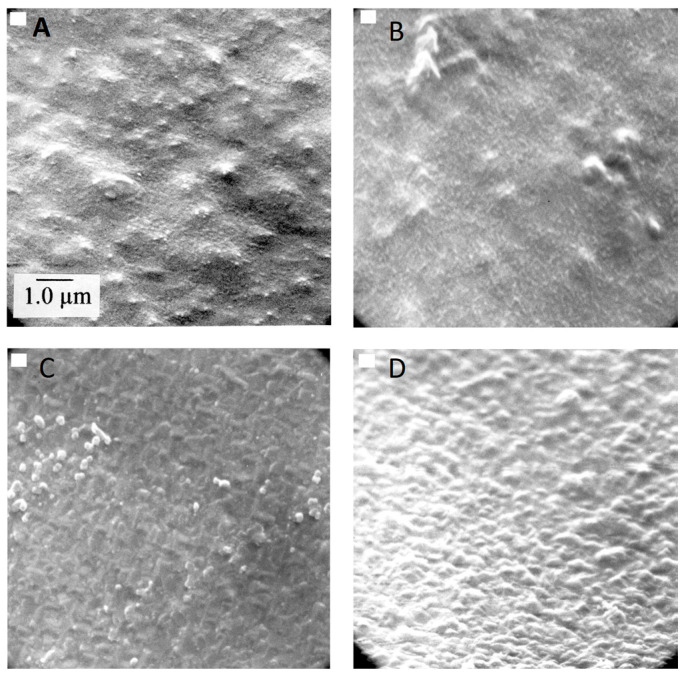


**Deposition chamber pressure:** Several studies have suggested that an increase of pressure favors the homogenous reactions resulting in higher growth rates and a larger crystallite size and thus in a rougher surface [26,111,116,152]. Other studies have shown an inversely proportional relationship between pressure and roughness [112,117,145], a decrease of the corresponding growth rate with smaller crystallite size and thus a smaller surface roughness. The reasons behind this discrepancy are not fully clear but we believe that the pressure effect primarily is specific to the chosen chemical system, the residence time, and the geometry of the reactor.**N-Doping:** N-doping studies showed that nitrogen doping with ammonia results in a slight increase of the surface roughness [19,25,26,118]. Quite probably, the high nitrogen concentration reduces the number of free sites on the growing surface due to the impurity surface coverage [124] leading to fewer but larger grains, thereby increasing the surface roughness. In addition, the possible deposition of Si_3_N_4_ at low temperatures (800–900 °C) could modify the surface roughness [19,26,118].**Hydrogen flow rate:** Hydrogen flow rate appears to have an impact on the surface roughness. A study has reported that an increase of hydrogen flow rate, while keeping the other precursors flow rates constant, decreases the average surface roughness significantly, improving the uniformity of the film [156]. Despite the lack of an explanation by this report, the roughness decrease could be attributed to an etching effect at a high H_2(g)_ flow rate, or a dilution effect resulting in modified partial pressures and hence the growth kinetics.

#### 4.1.3. Preferential Orientation

Experimental results have shown that the preferential orientation and the texture coefficient of polycrystalline SiC films can be modulated, according to the growth conditions [109,138].

**Growth Temperature:** Preferential orientation is strongly linked with the deposition temperature, which is the first order parameter (Figure 22). For temperatures lower than 1300 °C, it has been reported that the preferential orientation is the (111) orientation [81,109,137], while higher temperatures favor the (220) instead [8,81,82,109,110,111,137,139,157].
Figure 22Texture coefficient (calculated by the Harris model) of (111) (**a**) and (220) (**b**) planes between 1100 °C and 1500 °C (**a**) for various studies (References for this graph: Gavalas et al. (2025) [10], Gallou et al. (2003) [8], Choi et al. (1992) [81], Kim et al. (1995) [82], So and Chun (1988) [111]). Texture coefficient > 1, means that the corresponding plane is over-represented and when <1 is under-represented, compared to a randomly oriented deposit. Note that the texture coefficient absolute values depend on the number of XRD peaks considered in its calculation (higher number of peaks gives higher texture coefficient value). Note also that the operating parameters (pressure, flow rates, reactor geometry) were different for each study but kept constant when the deposition temperature was modified.
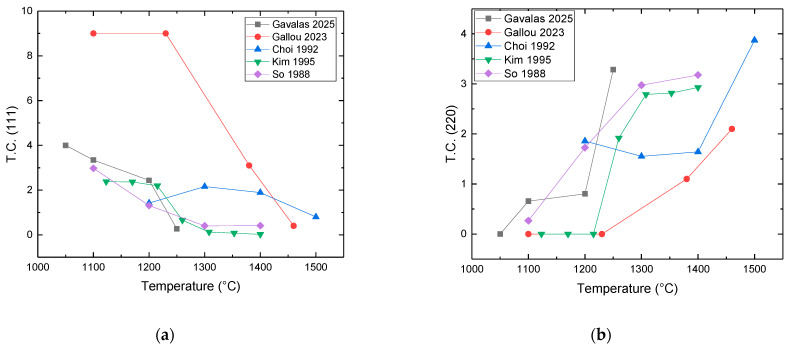


**(C/Si)_(g)_:** The increase of the (C/Si)_(g)_ ratio favors the (111) orientation independent of the deposition temperature (Figure 23) [81,95,109]. The authors of [81] believe that this is due to the steric hindrance effect. The latter is a slowing of chemical interactions due to non-bonding interactions. However, they notice that this effect is not clear and needs further study.
Figure 23Texture coefficient (calculated by Harris model) of (111) and (220) planes between 1200 °C and 1500 °C for (**a**) methyltrichlorosilane (C/Si)_(g)_ = 1 and (**b**) methyltrichlorosilane and propane (C/Si)_(g)_ = 2 [81]. Reprinted from “The Effects of C_3_H_8_ on the chemical vapor deposition of silicon carbide in the CH_3_SiCl_3_ + H_2_ System”, B.J. Choi, S.H. Jeun, D.R. Kim, *Journal of the European Ceramic Society*, Vol. 9, Pages No. 357–363, Copyright (1992), with permission from Elsevier.
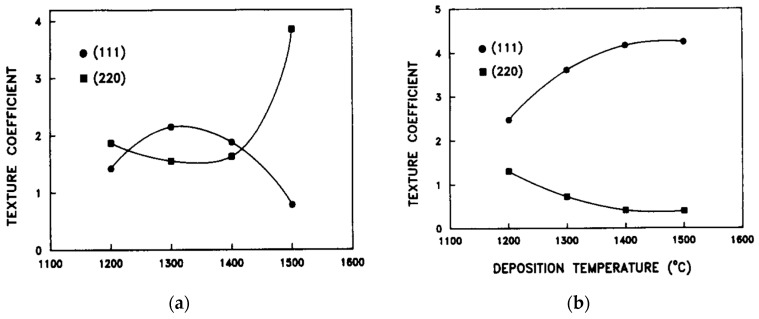


**Deposition chamber pressure:** It has been reported in one study [110] that an increase in pressure (from 1 kPa to 40 kPa) for polycrystalline SiC grown at 1400 °C favors the (111) in comparison to the (220) and (311) (Figure 24). Similar results are also shown in [139]. By contrast, So and Chun [111] show that an increase of pressure favors the (220} as the preferential orientation, instead of the (111} for films grown at 1300 °C. Once again, the effect of pressure depends on multiple parameters (gas flow/residence time, kinetics, diffusion coefficients) that can be different in the reported studies and thus result in the opposite behavior.
Figure 24(**a**) XRD diffractograms of polycrystalline SiC grown at 1400 °C showing the evolution of preferential orientation from (220) to (111) with increasing pressure [110]. (**b**) Evolution of (111), (200), (220), and (311) texture coefficients of polycrystalline SiC, grown at 1300 °C, with deposition pressure (data taken from Ref. [111]).
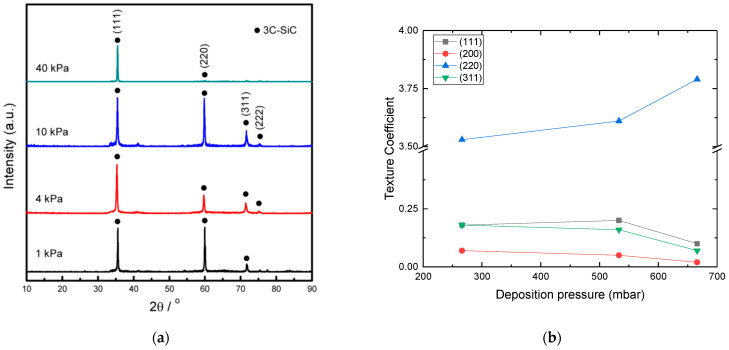


**Substrate:** The substrate orientation has a lesser effect in the preferential orientation of polycrystalline SiC for low temperatures (T < 1300 °C) as growth on substrates like the Si (100) ([72,81]), amorphous SiO_2_ substrates ([25,79]), or polycrystalline graphite gave similar results. For high temperatures, usually higher than 1500 °C, the adsorbed atoms have sufficient higher mobility and, then, the orientation of the chosen substrate can have a strong impact in the resulted grown orientation, shifting to hetero-epitaxial growth.**Residence time:** A way to control the residence time is to modify the substrate distance from the inlet. An increase of this distance increases the residence time. Kajikawa et al. [138] and Kim et al. [82] suggest that an increased residence time could change preferential orientation due to a change of the growing regime, as explained just below.

**Physical mechanisms for preferential orientation**. The preferential orientation of the polycrystalline deposited film can be modeled by the theory of van der Drift [158]. According to this theory, the preferential orientation of the polycrystalline deposited film, results from the orientation of the crystallites with the highest vertical growth rate. Even though this theory explains the preferential orientation in polycrystalline materials, the physical reasons behind the fast growth of specific orientation grains are not clear yet, and this also applies in the case of polycrystalline SiC. The following theories considering the surface energy [159], the growth regime [138], and twinning-mediated growth [8] have been proposed to address this issue.

**Surface Energy**: According to this theory, the plane with the lower surface energy will grow preferentially [159]. In general, the surface energy increases in order of (111), (100), (110) for the FCC lattice [159,160]. Nevertheless, Lee [160] has proposed that the selection of a low- or high-energy plane surface energy depends on the process conditions. For instance, a higher temperature favors the growth of a higher surface energy plane. Applying his conclusions in the case of SiC, one can say that for low temperature (mostly <1300 °C) the lowest energy plane is the (111), which grows preferentially, whereas at a higher temperature (>1300 °C) the higher energy (110) plane is favored.

**Growth regime:** According to Kajikawa [138] the change from (111) to (110) in the case of polycrystalline SiC growth, could be related to the competition between surface reaction rate and adsorption rate (sticking coefficient). Both rates are influenced by the concentration and the nature of the reactive species, hence by the residence time. Therefore, the preferential orientation will depend on the growth conditions. For example, an increase of substrate distance away from the inlet while keeping the deposition temperature constant increases the residence time of the gas phase resulting in a transition from (111) to (110) orientation for the polycrystalline SiC deposit [82,138]. The authors state that, whenever the growth is limited by the surface reaction rate, the resulting orientation is the (111), while there is a transition to the (110) for an adsorption limited growth, where the growth is limited by the sticking coefficient of the plane.

**Growth based on twinning:** The authors of [8] explain the origin of the (110) orientation by a twinning-mediated growth as the grown samples contained multiple twinned structure work (Figure 18a and Figure 25). It has been shown that this mechanism is activated at a high temperature and low supersaturation, both resulting in an enhancement of the adatoms surface mobility and a lower nucleation frequency. Additional experimental evidence of this mechanism is given by [161]. Several studies [119,137,157], although not explicitly mentioned, also show a typical of multiple twinned structure, suggesting as well that the transition to the (220) orientation is due to a twinning-based growth mechanism.
Figure 25(**a**–**c**) EBSD images of projected grains. IPF maps of (**d**–**f**) corresponding to the above images in order. They show fast-growing pentagonal multiple cross-twinned structure (see ref [8] for more details). Reprint by “Evidence of twin mediated growth in the CVD of polycrystalline silicon carbide”, Y. Gallou, M. Dubois, A. Potier, D. Chaussende, *Acta Materialia*, Vol. 259, Pages No. 119274, Copyright (2023), with permission from Elsevier.
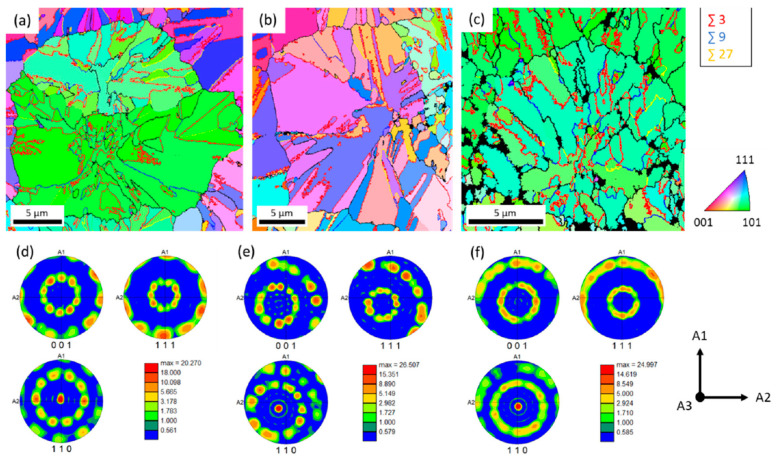


### 4.2. Residual Stress

The residual stress in thin films is a result of a combination of two types of stress: the intrinsic stress and the thermal stress [162,163].

#### 4.2.1. Intrinsic Stress

This is the stress that originates from the CVD growth process (growth stress). In polycrystalline thin films the intrinsic stress originates from three mechanisms. The first one originates from cluster coalescence resulting in a grain boundary area reduction and thus in tensile stress [164]. A second mechanism is related to the incorporation of additional atoms (like impurities but not limited) in the grain boundaries and induces compressive stress [162,165]. A third origin of intrinsic stress is related to the incorporation of dopant atoms in the crystalline lattice [102].

#### 4.2.2. Thermal Stress

The thermal stress is the stress that is formed during the cooling process after the CVD growth, due to different thermal expansion coefficients (CTE) between the deposit and the substrate [102]. The mean CTE value for polycrystalline 3C-SiC is reported to be equal to 2.47·10^−6^ °C ^−1^ according to [16]. In addition, a list of measured CTE values for β-SiC for a broad range of temperatures (25 K–3000 K) are available in [166]. Traditionally, the stress of a film is measured at room temperature. Thus, the corresponding cooling temperature equals the room temperature. Obviously, the thermal stress is proportional to the difference in growth temperature and room temperature. Often, the thermal stress is calculated by considering a constant value of CTE, not necessarily corresponding to the average CTE over the right temperature range. In the case of [112,150] calculating the thermal stress in polycrystalline SiC films, the mean value CTE_SiC_ (RT-800 °C): 4.625·10^−6^ °C^−1^ reported in [167] and CTE_SiC_ (950–1050 °C) = 4.63·10^−6^ °C^−1^ reported in [166], have been used, respectively. More accurate calculations of thermal stress use CTE as a function of deposition temperature [168]. A polynomial expression with order greater to 2 can be used to fit the curve CTE = f(T) [169,170]. This approach has been adopted by the authors of [10] for the case of polycrystalline SiC.

#### 4.2.3. Residual Stress Dependence on the Growth Conditions

The following analysis focuses on the intrinsic stress in polycrystalline SiC films, which could be originated from the density of the grain boundaries, the doping nature and the concentration as well as diffusion of adatoms in the grain boundaries [162].

**Growth Temperature:** In the case of the undoped polycrystalline growth of SiC on Si, the increase of deposition temperature (ranging from 700 °C to 850 °C) was shown to result in a lower residual stress whether the stress is tensile (positive value) [112] (Figure 26a) or compressive (negative value) [150] (Figure 26b). A detailed physical explanation has not been given in these reports. A possible origin for the lower tensile residual stress when the increase of temperature increases (Figure 26a) is the subsequent increased crystallite size and hence the reduction of the grain boundary density [164,170]. In the second case (Figure 26b), the authors suggest that surface stress is the source of the observed compressive residual stress, which is important only in the early stages of growth [150]. The origin of compressive stress could be better explained by the diffusion of adatoms in the grain boundaries as proposed by Chason [171]. Moreover, the complex behavior observed in Figure 26b can be interpreted from the competition between tensile stress naturally induced by the grain boundaries and compressive stress induced by extra atom insertion at the grain boundaries. This competition is ruled by surface kinetics and depends on multiple factors related to growth parameters (grain size, diffusion coefficient, adatoms concentration).In the case of nitrogen-doped polycrystalline films the residual stress is independent of the deposition temperature, according to [10]. The authors state that this independency is due to compensating effects of deposition temperature and nitrogen incorporation on the intrinsic stress.
Figure 26Growth temperature dependence of measured at room temperature residual stress for polycrystalline SiC deposited on Si substrate (**a**) from [112] and (**b**) from [150]. The thermal stress is calculated from CTE difference, and the intrinsic stress is extracted by subtracting from the measured residual stress the calculated thermal stress.
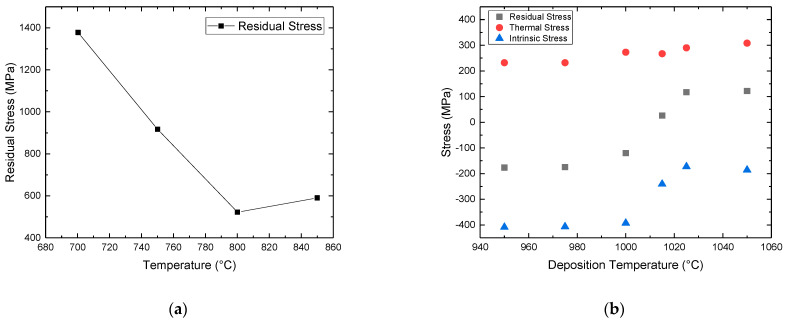


**Deposition chamber pressure:** Pressure also affects the residual stress of undoped [112,143,152,172] and n-doped films [26,145,146]. For the undoped films, both an increase [112] and a decrease [143,152,172] of the tensile residual stress has been observed. Indeed, the pressure effects can be positive or negative depending on the precursors’ system, possible dopant atoms, and the growth conditions (Figure 27). In the n-doped case, an increase of the pressure resulted in a lower tensile residual stress [26,145,146] (see Figure 27). The origin of this is not yet understood and needs to be further investigated. However, as we showed earlier, pressure can have a positive or a negative effect on the grain size, the observed behavior in terms of stress can also be related to the grain size.
Figure 27Residual stress vs. deposition pressure for undoped polycrystalline SiC (Red) (References for this graph: Fu et al. (2004) [143], Liu et al. (2009) [112], Fu et al. (2005) (1) [152], Fu et al. (2005) (2) [172]) and n-doped polycrystalline SiC (Black) (Refs: Fu et al. (2014) [26], Trevino et al. (2005) [145], Trevino et al. (2014) [146]). Note also that the operating parameters (flow rates, temperature, reactor geometry) were different for each study but kept constant when the pressure was modified.
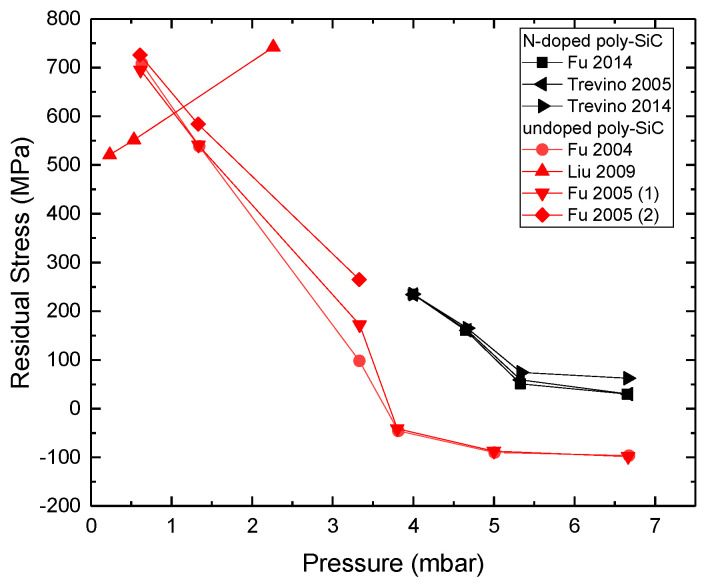


**(C/Si)_(g)_:** Several works have shown that in the case of CVD deposition with DCS, the (C/Si)_(g)_ ratio affects the residual stress of the films deposited on Si and SiO_2_ substrates [25,112,142,144,147] (Figure 28a). According to these studies, the decrease of (C/Si)_(g)_ results in co-deposition of Si_(s)_ which reduces the tensile residual stress as in a SiC_(s)_ + Si_(s)_ film where the average bond length is bigger compared to the average bond length of stoichiometric SiC_(s)_. Indeed, a model has been proposed in the case of silicon nitride growth according to which a higher amount of Si–Si bonds within the crystalline lattice result in a lower tensile residual stress since the Si–Si covalent bond is longer than the Si–N bond [173].**(Cl/Si)_(g)_:** To our knowledge there are no reports in the literature on the effect of chloride to the residual stress. However, as we have discussed previously that the crystallite size is proportional to (Cl/Si)_(g)_, and by considering the grain coalescence model [164] and the incorporation of extra atoms at the grain boundaries, it would be expected that the intrinsic residual stress would also be modified.**N-Doping:** Nitrogen doping with ammonia increases the tensile stress proportionally to the ammonia flow rate (Figure 28b) [19,25,146,148]. The Si–N bond length is smaller than the Si–C bond, resulting in a lattice shrinkage and hence a higher tensile stress similar to the observations in the n-doped epitaxial 4H-SiC [174,175].
Figure 28Dependence of residual stress of polycrystalline SiC films: (**a**) on DCS flow rate while other process parameters kept constant. (References for this graph: Liu et al. (2009) [112], Liu et al. (2010) [25], Fu et al. (2011) [144], Roper et al. (2008) [147]) (**b**) on NH_3(g)_ flow rate with other process parameters are kept constant (References for this graph: Fu et al. (2007) [19], Liu et al. (2010) [25], Fu et al. (2014) [146], Roper et al. (2009) [148]) Note that the operating parameters (pressure, total flow rate, temperature, reactor geometry) were different for each study but kept constant when the flow rates were modified.
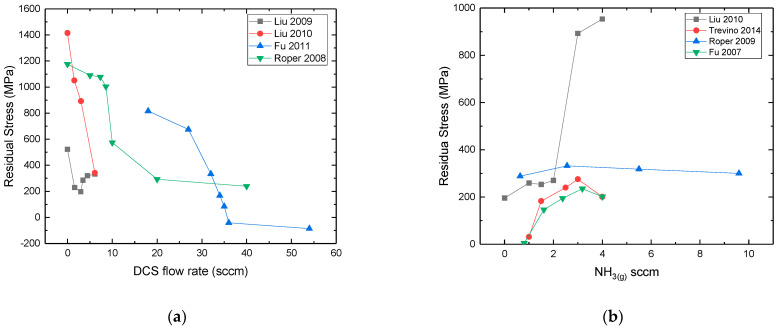


**Annealing:** Post-deposition-annealing in the range of 900–1100 °C of n-doped SiC grown at 800 °C, lowers the residual tensile stress compared to the as-grown stress [176]. A shift of nitrogen atoms from the weaker N–C bonds to the formation of more stable Si–N bonds, could increase the average bond length and decrease the corresponding residual stress, since the Si–N bond is longer than N–C [25,175]. However, this study [176] does not consider the eventual thermal stress due to the annealing. Additional calculations are required to de-couple the effects of annealing on the intrinsic and thermal stress.

### 4.3. Electrical Resistivity

Polycrystalline SiC resistivity has been modulated intentionally mainly by n-type doping [10,118,119,125,126,127,128]. In addition to the dopant activation, the Seto model [177] has been evoked in many experimental works related to the resistivity of polycrystalline SiC [31,119,125,126,148]. According to this model proposed for polycrystalline Si films, the resistivity is inversely proportional to the crystallite size and the corresponding value of the potential barrier for the trapping of conductive carriers has been determined in the case of polycrystalline SiC (0.0402–0.29 eV) [125] and (0.17 eV) [31].

**Dopant agent flow rate:** Several studies [19,78,118,136,148,149] show that an increased ammonia flow rate drops the resistivity by several orders of magnitude due obviously to an increasing doping of the polycrystalline SiC. After a certain level of ammonia flow rate, the resistivity reaches a plateau value, with the lowest resistivity found around 0.005 Ω·cm in [19], which indicates the doping limit of the polycrystalline SiC (Figure 29). Reaching the solubility limit and/or the ion scattering mechanism due to high dopant concentration as well as a possible deposition of Si_3_N_4_ could explain the limitation in the resistivity drop (Figure 29) [25,118]. Note that all the studies dealing with the nitrogen doping of polycrystalline SiC have been performed at relatively low temperatures (800–1000 °C). N-doping is also possible by phosphorus using PH_3(g)_ as shown in [104,149]. In these studies, the resistivity decreases with the flow rate of PH_3(g)_ and reaches a lower limit value of 0.02 Ω·cm.
Figure 29Room temperature electrical resistivity of polycrystalline SiC vs. dopant agent flow rate. In parenthesis the deposition temperature for each case is indicated (References for this graph: Wijesundara et al. (2002) [118] and (2003) [78]), Liu et al. (2010) [25], Fu et al. (2007) [19], Noh et al. (2007) [136], Roper et al. (2009) [148], Furumura et al. (1988) [149]). Note that the operating parameters (pressure, flow rates, temperature, reactor geometry) were different for each study but kept constant when the NH_3_ flow rate was modified.
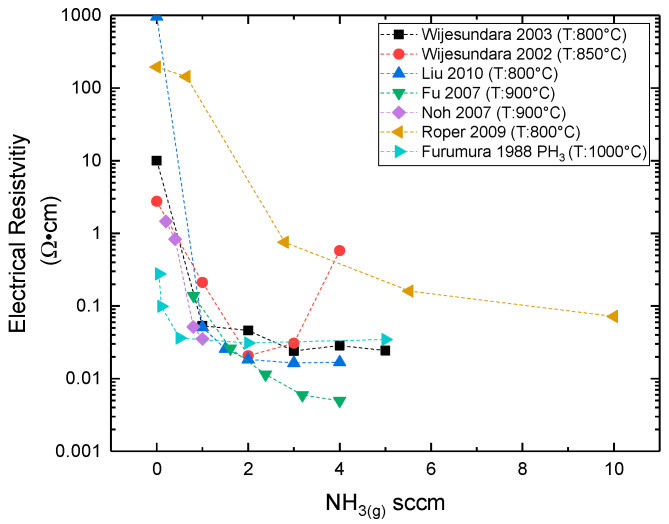


**Growth Temperature:** An increase in deposition temperature decreases the corresponding resistivity (Figure 30) [10,31,78,107]. Most authors propose that the decreased resistivity is related with an improved film crystallinity [78,107] and possibly to an increase in nitrogen incorporation at higher deposition temperatures [178]. Another explanation is that the increase in deposition temperature results in an increase of the grain size (or an equivalent decrease of the grain boundaries density) and thus in a decrease of the resistivity according to the Seto model [177] relating to the resistivity with grain size.
Figure 30Deposition temperature effect on room temperature electrical resistivity of n-doped polycrystalline SiC (References: Wijesundara et al. (2003) [78], Cheng et al. (2002) [107], Privitera et al. (2015) [31], Gavalas et al. (2025) [10]). Note that the operating parameters (pressure, flow rates, reactor geometry) were different for each study but kept constant when the deposition temperature was modified.
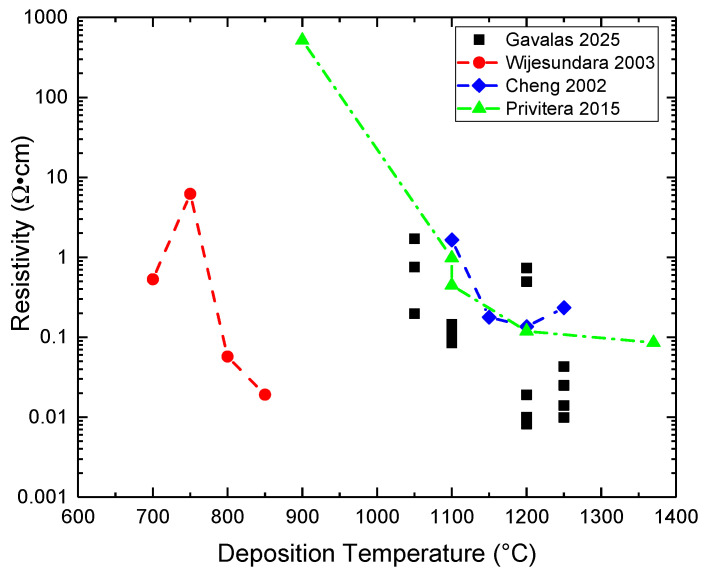


**Deposition chamber pressure:** According to [19,26], the increase of pressure results in a slight increase of resistivity for polycrystalline SiC doped with nitrogen by ammonia flow. The authors propose that the increase of pressure may result in the formation of fewer Si–N bonds within the lattice and more nitrogen atoms in vacancies, which are electrically inactive and do not contribute to the doping effect. However, there is no evidence to support this explanation. Anyway, the pressure effect is weak because it causes very small changes in the resistivity of the polycrystalline SiC film.**(C/Si)_(g)_**: According to [25], a decrease of the (C/Si)_(g)_ ratio results in a decrease of resistivity in the case of polycrystalline SiC growth with DCS flow rate and in situ doping with ammonia. The authors propose as a possible explanation an increase in nitrogen incorporation due to a doping by site competition [130]. Indeed, since carbon and nitrogen compete for the same site in SiC, a decrease in (C/Si)_(g)_ favors the incorporation of N in SiC.**(Cl/Si)_(g)_:** To our knowledge, only the work of Liu et al. [25] indirectly reports the change of electrical resistivity with the (Cl/Si)_(g)_ ratio. However, due to the employed precursors, there is a simultaneous variation of the (C/Si)_(g)_ and, based on that variation, the authors propose the site competition theory as an explanation for the variation of resistivity. Nevertheless, it is known that by increasing the Cl_2(g)_ flow rate, the amount of deposited Si_(s)_ clusters in the n-doped polycrystalline SiC is suppressed on one hand and the crystallite/grain size is also increased on the other. In the latter case, a decrease of the resistivity is expected according to the Seto model. Obviously, a study on the single effect of (Cl/Si)_(g)_ on the electrical resistivity of undoped or doped polycrystalline SiC is currently missing from the literature.**Thickness/Deposition Time:** Thickness dependent resistivity (by increasing the deposition time) studies have also been conducted [126]. The resistivity of n-doped polycrystalline SiC decreases with the deposited thickness as thicker films have larger grain and crystallite size, resulting in less carrier trapping, as discussed by Seto. An in-depth variation of doping has been partly excluded from the constant in-depth nitrogen concentration according to SIMS measurements [126].**Annealing:** The resistivity of polycrystalline SiC was found to be inversely proportional with the annealing temperature, the latter being greater than the deposition temperature (Figure 31) [78,148]. During annealing at T ≥ 900 °C, a possible shift of interstitial nitrogen from the N–C bonds atoms to the Si–N bonds, activates the doping mechanism of the nitrogen atoms, thus decreasing the resistivity. Therefore, post-deposition annealing helps to improve the conductivity of n-doped films grown at relatively low temperatures.
Figure 31The resistivity of nitrogen doped polycrystalline SiC films with the annealing temperature. (References: (Black) Wijesundara et al. (2003) [78], (Red) Roper et al. (2009) [148]). The triangle data show the corresponding resistivity value of the deposition temperature.
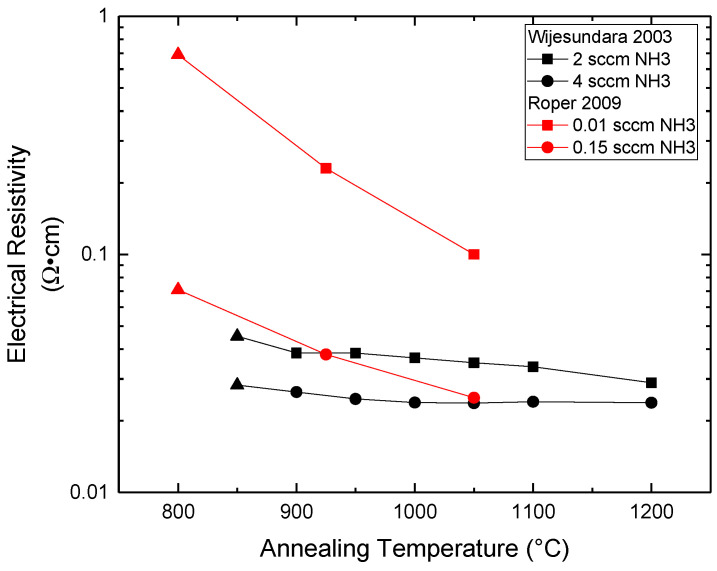


### 4.4. Thermal Conductivity

In this part, the effect of microstructure and composition on thermal conductivity is discussed in agreement with the related literature.

**Grain size and crystalline quality:** In polycrystalline SiC deposited by CVD, the thermal conductivity is shown to be inversely correlated to the grain size, hence density of the grain boundaries, due to their scattering effect on the phonon [21]. In other words, the greater the grain size, equivalently the lower the grain boundary density, the higher the thermal conductivity. This scattering by the grain boundaries was shown to be the dominating scattering mechanism at a low temperature (below 200 K) [21]. In [179], multiscale modeling involving molecular dynamics and finite element methods confirmed the detrimental effect of the grain boundaries on thermal conductivity and showed good trend compared to the experiments made in [21], although the drop of thermal conductivity was observed to appear for much smaller grains in the modeling, likely due to oversimplification of the structure of the grain boundaries. The effect of the grain boundaries can also be observed when comparing the thermal conductivities parallel and perpendicular to the surface of the columnar deposit. Parallel to the surface of the deposit, the phonon will cross more grain boundaries than perpendicular to it, due to the columnar nature of the grains, decreasing the apparent thermal conductivity. In [180], the decrease of thermal conductivity for the direction parallel to the surface of the deposit was measured to be 15%. A similar effect was observed in CVD diamond deposits [181]. A correlation between the increase of stacking faults along the (111) direction and the decrease of thermal conductivity was also established in [180], suggesting that stacking faults also act as thermal resistance. As discussed earlier, an increase of temperature can be related to an increase of crystalline quality (fewer stacking faults) and an increase of the grain size (fewer grain boundaries), with both resulting in higher thermal conductivity. Other parameters having an influence on the grain size and crystalline quality (pressure, (C/Si)_(g)_ ratio, (Cl/Si)_(g)_ ratio) would then also result in variation of thermal conductivity of the corresponding polycrystalline SiC deposits.**Doping**: Doping has a detrimental effect on thermal conductivity due to phonon-charge carrier interaction. In ref [22], for instance, non-intentionally doped polycrystalline 3C-SiC deposited by CVD (CH_3_SiCl_3_, 1450–1500 °C) was shown to exhibit higher thermal conductivity than nitrogen and nitrogen-boron doped SiC. The decrease of thermal conductivity associated with the increase of charge carrier density is predicted to be higher for holes (p-type) than for electrons (n-type) [179]. Therefore, growth conditions that favor doping of SiC (increase of doping agent mass flow, temperature) would result in a decrease of thermal conductivity.

## 5. Conclusions

Following are summarized the most important conclusions extracted from the above bibliographic study.


Structure and growth


Polycrystalline SiC thin films can grow in the temperature range of 800–1500 °C, either with the C–H–Si or with the C–Cl–H–Si systems, in various ranges of pressure and precursors flow rates. Substrates, such as Si (100), SiO_2_/Si(100), Al_2_O_3_, and graphite, can be used for the deposition of polycrystalline SiC. For temperature and (C/Si)_(g)_, the (Cl/Si)_(g)_ ratios seems to be the most important factor for the deposition, as these ratios control the deposition rate, crystallinity, preferential orientation, chemical composition, and the surface roughness. (C/Si)_(g)_ and (N/C)_(g)_ control the doping efficiency of the deposited material. The pressure effect on chemical composition, deposition rate, grain size, surface roughness, and residual stress varies depending on the selected precursors and the growth conditions, hence drawing a conclusion from multiple studies is not straightforward. Silicon deposition can be avoided with the addition of chlorine to the C–H–Si system, the dominating factor being the (Cl/Si)_(g)_ ratio, due to the high affinity of chlorine with silicon. In turn, (Cl/Si)_(g)_ also affects positively the crystallite size of polycrystalline SiC but negatively its growth rate, along with the surface roughness, possibly due to the surface poisoning effect of Cl.


Residual Stress


In most cases, a tensile residual stress has been reported for polycrystalline SiC. Residual stress originates from the intrinsic stress and thermal stress. Thermal stress depends on the CTE of the grown film and the substrate, as well as the difference between deposition temperature and room temperature *ΔΤ*. The higher the CTE difference and/or the *ΔΤ* the higher the value of thermal stress. Intrinsic stress complexity, however, is higher due to the multiple mechanisms involved (cluster coalescence, diffusion of adatoms in the grain boundaries, surface stress, and doping). The intrinsic stress is affected by temperature. The intrinsic stress can also be affected by the (C/Si)_(g)_, (Cl/Si)_(g)_, and (N/Si)_(g)_ ratios, and therefore by the stoichiometry and the doping of the grown material. Higher incorporation of Si_(s)_ shifts the growth stress towards less tensile/more compressive values due to an increase of the average bond length (Si–Si bonds are longer than Si–C bonds). By contrast, the higher incorporation of N_(s)_ impurities shifts the growth stress towards more tensile values due to the shorter value for the Si–N bond compared to the Si–C bond. The effect of chlorine has not been sufficiently characterized but, as it was shown to induce change of the crystallite size, it could be expected to modify the intrinsic stress.


Electrical Resistivity (N-doping)


Nitrogen and phosphorous results in n-doping to the polycrystalline SiC thin films, as giving an excess electron to the system with each nitrogen or phosphorous atom that it is introduced into the SiC lattice. Ammonia has been proven to be a good precursor for nitrogen doping, better than N_2(g)_ due to its higher reactivity. The n-doped films exhibit resistivity down to 5 mΩ·cm. Electrical resistivity is inversely proportional to ammonia flow rate. However, the resistivity drop reaches a limited value, probably due to the solubility limit of the nitrogen atoms in SiC, the reduced electron mobility due to ion scattering in high dopant concentration or possibly due to a small deposition of Si_3_N_4(s)_ for high flow rates of ammonia. Deposition temperature was found to reduce the resistivity of the corresponding SiC deposit, due to the enhancement of the crystallinity of the deposited films. Indeed, the resistivity is expected to be inversely proportional to the crystallite size in doped films (especially the light-doped films) according to the Seto model, which is invoked by many experimental studies in the case of polycrystalline SiC. This is also why resistivity can be inversely proportional to the film’s thickness, as a higher thickness results in a greater grain size, resulting in less carrier trapping by the grain boundaries. In some studies, chamber pressure was found to increase slightly the resistivity of the grown polycrystalline SiC, which was interpreted as a higher amount of electrically inactive N atoms resulting from an increase of pressure. A decrease of the (C/Si)_(g)_ is shown to result in a higher nitrogen incorporation in the polycrystalline SiC deposit due to site competition which in turn decreases the electrical resistivity.

## 6. Future Perspectives

According to the literature results on the polycrystalline SiC LPCVD growth, only the deposition temperature effect has been understood in some depth. Despite the large number of results given in this review, the effect of the other growth conditions is not yet fully understood. This is due to the complexity of the LPCVD technique, and the growth mechanisms involved (homogeneous and surface reactions). It appears that the effect of deposition on the polycrystalline film’s properties can vary from one study to another, depending on the chemical system and the growth conditions used. Future studies on the pressure effect, first on the microstructure and then on the resulting physical properties (stress, doping efficiency, electrical resistivity) would be beneficial. The chloride-induced effects (especially on the residual stress and the electrical resistivity, both in un-doped and n-doped polycrystalline SiC) also require systematic and in-depth study. Note that there are numerous papers reporting the use of different chlorinated precursors or additional chlorine-source gases, but the results are not comprehensive in terms of variation of process conditions and/or resulting material properties. Finally, additional work on the nitrogen doping of polycrystalline SiC is required to have a better understanding of the NH_3(g)_ and N_2(g)_ on the corresponding film microstructure and its properties. Overall, we believe that process modeling, together with experimental work, is crucial to better relate local growth conditions to the composition and microstructure of polycrystalline SiC deposits as these parameters have been shown to control the final properties of interest.

## Figures and Tables

**Table 4 micromachines-16-00276-t004:** Bibliographic polycrystalline SiC CVD growth conditions and corresponding properties with the C–H–Si chemistry.

T (^°^C)	P (mbar)	Si Source	C Source	H_2_ Sccm	NH_3_ Sccm	XRD Peaks	ρ (Ω·cm)	Stress (MPa)	Reference	Solid Composition (Meas. Methods)	Remarks (Meas. Methods)
900–1400	0.40	SiH_4_	C_3_H_8_	850	-	(111),(200)	-	-	[105]	SiC_(s)_ (XRD) (C/Si)_(s)_: 1.2 (XPS)	T, P, and FR study on structure (XRD, SEM, XPS)
800–1000	0.001	DSB	DSB	-	-	(111),(200),(220),(004)	-	-	[131]	SiC_(s)_ (XRD)	T study on structure. Chemical stability study (XRD, SEM, AFM)
850–900	0.25–0.55	DSB	DSB	-	-	(111),(200),(220),(311), (222)	-	-	[106]	SiC_(s)_ (XRD) (C/Si)_(s)_~1 (XPS)	Structure vs. T (XRD, XPS, SEM)
1300	1013	SiH_4_	C_3_H_8_	25,000	-	(111),(200),(220)	-	-	[72]	SiC_(s)_ (XRD)	Structure vs. substrate (XRD, SEM, XTEM)
700–900	0.013–2	MS	MS	-	PH3: 10	-	-	-	[104]	SiC_(s)_ (FTIR) (C/Si)_(s)_~1 (XPS)	T and P study on structure and doping (TEM, XPS, FTIR, SIMS)
900–1150	0.66–2	TMS	TMS	1000–2000	-	(111)	-	-	[132]	(C/Si)_(s)_ < 1 (EPMA), T: 900–980 °C (C/Si)_(s)_~1 (EPMA), T: 1000–1150 °C	T study on structure and orientation (XRD, XPS, SEM, TEM, EPMA)
600–1100	1.3	DSCB	DSCB	-	-	(111),(200),(220),(311)	-	-	[108]	SiC_(s)_ (XRD) SiC_(s)_ (FTIR)(Auger) (C/Si)_(s)_: 0.92, T: 800 °C (C/Si)_(s)_: 1, T > 900 °C	T study on structure (XRD, SEM, Auger, FTIR)
800	0.13	DBS	DBS	-	1–4	(111)	0.03	-	[125]	SiC_(s)_ (XRD)	NH_3_ FR study on structure, resistivity and doping (XRD, SIMS, 4PP, VdP, Hall)
650–850	0.03–0.06	DBS	DBS	-	0–5	(111),(200),(220)	0.02	-	[78]	SiC_(s)_ (XRD) (C/Si)_(s)_~1 (XPS)	T and NH_3_ FR study on structure and electrical properties (XRD, XPS, 4PP, Post Annealing)
850	0.01	DBS	DBS	-	0–5	(111),(200),(220),(311)	0.02	-	[118]	SiC_(s)_ (XRD) (C/Si)_(s)_~1 (XPS) (C/Si)_(s)_~1 (Auger)	NH_3_ FR study structure, growth rate and electrical properties (XRD, XPS, Auger, 4PP)
850	0.01	DBS	DBS		-	(111),(220)	-	-	[133]	SiC_(s)_ (XRD)	Study on structure (XRD, SEM, TEM)
1100	53.0	SiH_4_	C_3_H_8_	3000	0–5	(111)	<10^−3^	-	[79]	SiC_(s)_ (XRD) (C/Si)_(s)_~1 (XPS)	Doping study (NH_3_) (XRD, XPS, SIMS, vdP)
1800 (365)	5.30	SiH_4_	CH_4_	200–1000	N_2_ 0–150	(111),(220),(311)	0.2	-	[128]	SiC_(s)_ (XRD) (C/Si)_(s)_~1 (XPS): undoped (C/Si)_(s)_ < 1 (XPS): n-doped	HWCVD: N_2_, H_2_ FR study on electrical and structural properties (XRD, XPS, vdP, Hall)
850–1050	0.04–1.05	HMDS	HMDS	20	-	(111),(200),(220),(311)	-	-	[134]	SiC_(s)_ (XRD) (C/Si)_(s)_~1 (Auger) (C/Si)_(s)_~1 (XPS) (EPMA) (C/Si)_(s)_ ≤ 1, T: 1050 °C (C/Si)_(s)_ > 1, T < 1050 °C	T study on structure and composition (XRD, SEM, Auger, EPMA, XPS)
1600 (360)	-	MMS	MMS	200	TMA: 0–0.033	(111)	10^3^	-	[20]	SiC_(s)_ (XRD)	HWCVD: TMA FR study on structure and conductivity (XRD, TEM, FTIR, SIMS, conductivity)
1200–1250	-	HMDSO	HMDSO/C_6_H_14_	0 or 600	-	(111),(200),(220),(311)	-	-	[135]	SiC_(s)_ (XRD) SiC_(s)_ + C_(s)_ (Raman) (XPS)-T (C/Si)_(s)_: 1.6, T: 1200 °C (C/Si)_(s)_: 8, T: 1250 °C (XPS)-(C/Si)_(g)_ (C/Si)_(s)_: 1.6, (C/Si)_(g)_: 6 (C/Si)_(s)_: 1.1, (C/Si)_(g)_: 9	Phase composition, bonding state, microstructure, hardness, and elastic modulus study by varying the precursor proportion mix of precursors (XRD, Raman, XPS, SEM, STEM)
850–1070	5	MS	MS	-	-	(111),(200),(220),(311)	-	-	[97]	SiC_(s)_ (XRD) SiC_(s)_ + C_(s)_ (Raman)	T and FR study on structure, preferential orientation and stacking faults (XRD, Raman, TEM)

DSB: 1,3-disilabutane; MS: methylsilane; DCS: dichlorosilane; TCS: trichlorosilane; DSCB: 1,3-disilacyclobutane; HMDS: hexamethyldisilane; MMS: monomethylsilane; TMA: trimethylaluminum; VTC: Vinyltrichlorosilane; XRD: X-Ray diffraction; vdP: Van der Pauw; SEM: scanning electron microscopy; XPS: X-Ray photoelectron spectroscopy; 4PP: 4-point probe; TEM: transmission electron microscopy; XTEM: X-Ray transmission electron microscopy; SIMS: secondary ion mass spectroscopy; AFM: atomic force microscopy; HWCVD: hot wire CVD; HLCVD: halide laser CVD; T: deposition temperature; P: deposition pressure, FR: flow rate.

**Table 5 micromachines-16-00276-t005:** Bibliographic polycrystalline SiC CVD growth conditions and corresponding properties with the C–Cl–H–Si chemistry.

T (°C)	P (mbar)	Si Source	C Source	H_2_ Sccm	NH_3_ Sccm	XRD Peaks	ρ (Ω·cm)	Stress (MPa)	Reference	Solid Composition (Meas. Methods)	Remarks/(Meas. Methods)
900	2.6	DCS	C_2_H_2_	-	0.2–1	(111)	0.036	-	[136]	SiC_(s)_ (XRD)	NH_3_ FR study on structure and electrical properties (XRD, 4PP)
1150–1600	86–666	MTS	MTS	250–1250	-	(111),(200),(220),(311)	-	-	[109]	SiC_(s)_ (XRD) (Microprobe) (C/Si)_(s)_ < 1, T < 1400 °C (C/Si)_(s)_~1, T: 1400 °C (C/Si)_(s)_ > 1, T > 1400 °C	T and P study on structure, surface morphology, and orientation (XRD, SEM, Microprobe)
1100–1400	266–666	MTS	MTS/C_3_H_8_	-	-	(111),(200),(220),(311)	-	-	[111]	SiC_(s)_ (XRD)	T, P, and MTS FR study on structure and surface morphology (XRD, SEM, TEM)
1200–1600	9300	MTS	MTS	-	-	(111),(220)	-	-	[137]	SiC_(s)_ (XRD)	T study on structure, surface morphology, and orientation (weight measurement, XRD, SEM)
1100–1250	18	TMS	TMS	-	N_2_ (0–10)	(111)	0.1	-	[107]	SiC_(s)_ (XRD)	T and TMS + N_2_ FRs investigation of structure and doping (XRD, SEM, AFM, vdP, Hall)
1000–1400	13–100	MTS	MTS	1000	-	(111),(200),(220),(311)	-	-	[82]	SiC_(s)_ (XRD) (C/Si)_(s)_~1 (Auger) (C/Si)_(s)_~1 (XPS)	T and P study on structure and preferential orientation (weight measurement, XRD, SEM, Auger, XPS)
1000–1300	Varies	MTS	MTS	-	-	(111),(220)	-	-	[138]	-	Model of controlling the preferential orientation
1200–1600	10–400	SiCl_4_	CH_4_	-	-	(111),(200),(220),(311)	-	-	[139]	SiC_(s)_ (XRD)	T and P study on structure and orientation (XRD, SEM)
1300–1600	10–400	SiCl_4_	CH_4_	-	-	(111),(200),(220),(311)	-	-	[110]	SiC_(s)_ (XRD)	T and P study on structure and orientation (XRD, SEM, TEM)
1000–1400	10–100	MTS	MTS	950	-	(111),(200),(220),(311)	-	-	[8]	SiC_(s)_ (XRD)	Reactor modeling: Τ effect on growth rate, orientation and structure (XRD, SEM, EBSD)
900–1250	20	SiH_4_	C_3_H_8_	400	1	(111),(200),(220),(311)	0.008	252	[10]	SiC_(s)_ (XRD)	Τ study on structure, orientation, surface morphology, stress and resistivity (SEM, XRD, Optical Profilometry, 4PP, Hall, Ellispometry)
750–1050	0.004	DSC	C_2_H_2_	500	-	(110), (111),	-	-	[140]	-	T study on structure (ECP, SEM)
800–1400	0.06	MTS/DSB	MTS/DSB	-	-	(111),(200),(220),(400)	-	-	[141]	SiC_(s)_ (XRD)	T study on structure, surface morphology, and defects (XRD, SEM, TEM)
750–850	0.22–2.2	MS/DCS	MS	240	-	(111),(200),(220)	-	300	[112]	SiC_(s)_ + Si_(s)_ (XRD)	T, P, and DCS FR study on stress (AFM, XRD, Reflectometry, curvature measurement, strain gauge)
800	0.23	MS/DCS	MS	240	0–5	(111),(200),(220)	0.016	350	[25]	SiC_(s)_ + Si_(s)_ (XRD)	NH_3_ + DCS FRs study on mechanical and electrical properties (XRD, SEM, 4PP, profilometry/curvature measurement/cantilevers)
900	5.3	DCS	C_2_H_2_	-	64	(111)	0.0054	-	[126]	-	Thickness effect on electrical properties (SIMS, vdP)
900	0.53–6.6	DCS	C_2_H_2_	-	-	(111)	-	172	[142]	SiC_(s)_ (XRD)	DCS FR study on structure and stress (SEM, TEM, XRD, curvature measurement)
900	0.53–6.6	DCS	C_2_H_2_	-	0.8–5	(111)	0.01	25	[19]	SiC_(s)_ (XRD)	NH_3_ FR study on resistivity and stress/strain gradient (XRD, SIMS, curvature measurement, cantilevers/lateral resonator 4PP)
900	0.53–6.6	DCS	C_2_H_2_	-	-	(111)	-	25	[143]	SiC_(s)_ (XRD)	P study on stress (XRD, SEM, TEM, curvature measurement)
1200–1500	46.6	MTS	CH_4_	4000	-	(111),(200),(220),(311)	-	-	[95]	SiC_(s)_ (XRD) (EPMA)-T -(C/Si)_(s)_:0.49, T: 1200 °C -(C/Si)_(s)_: 0.79, T: 1300 °C -(C/Si)_(s)_: 1.2, T: 1450 °C (EPMA)-CH_4_ -(C/Si)_(s)_: <1, (C/Si)_(g)_ < 2.2 -(C/Si)_(s)_~1, (C/Si)_(g)_ ≥ 2.2	T and CH_4_ FR study on the growth rate, structure, surface morphology, preferable orientation, and chemical composition (XRD, SEM, TEM, EPMA)
1000	1013	SiH_4_	C_2_H_4_	8000	-	(111),(220)	-	-	[96]	(Auger) for (Cl/Si)_(g)_: 0–50 -(C/Si)_(s)_: 0.4–0.9, (C/Si)_(g)_: 0.25 -(C/Si)_(s)_: 1.3–1.1, (C/Si)_(g)_: 1 -(C/Si)_(s)_: 1.5–1.2, (C/Si)_(g)_: 2	HCl FR study on structure (XRD, SEM, TEM, Auger)
952–1000	33	DCS	C_3_H_8_	6000	-	No data	-	-	[115]	(Auger) for (Cl/Si)_(g)_: 10–20 (C/Si)_(s)_: 0.7–0.9	HCl FR study on growth rate, stoichiometry, and structure (NOS, AES, TEM)
900	2.6	DCS	C_2_H_2_	-	-	(111),(220)	3	132	[144]	SiC_(s)_ (XRD)	DCS FR study on structure, stress, strain gradient, and resistivity (XRD, SEM, curvature measurement, 4PP, strain gauge, lateral resonator)
900	4–7	DCS	C_2_H_2_	-	3.2	(111),(220)	0.008	53	[145]	SiC_(s)_ (XRD) (C/Si)_(s)_~1 (SIMS) (C/Si)_(s)_~1 (XPS)	P study on growth rate, structure, doping, resistivity, strain gradient, and stress (XRD, SEM, curvature measurement, SIMS, AFM, XPS, 4PP, lateral resonator, cantilevers).
900	5.3	DCS	C_2_H_2_	-	3.2	(111),(220)	-	75	[146]	-	P and NH_3_ FR study on stress, strain gradient (Curvature measurements, cantilevers, stress pointers, lateral resonator)
800	0.19–0.24	DSB/DCS	DSB	-	-	(111),(220),(311)	-	250	[147]	SiC_(s)_ + Si_(s)_ (XRD) (EPMA) (C/Si)_(s)_: 1, (C/Si)_(g)_: 1 (C/Si)_(s)_: 0.83, (C/Si)_(g)_ < 1	DCS FR study on structure and stress (XRD, SEM, AFM, EPMA, optical reflectometry, curvature measurement)
800	0.23	DSB/DCS	DSB	240	3	(111),(220),(311)	0.08	300	[148]	(C/Si)_(s)_~1 (SIMS) (C/Si)_(s)_~1 (EPMA) (C/Si)_(s)_~1 (XPS)	NH_3_ FR study on resistivity, residual stress, and % atomic nitrogen in deposited film (4PP, SIMS, EPMA, XPS, curvature measurement)
1000	2	TCS	C_3_H_8_	7000	PH3 0.1–10	(111),(220)	0.03	-	[149]	SiC_(s)_ (XRD) (C/Si)_(s)_ > 1 (XPS)	C_3_H_8_ + PH_3_ FRs study on structure and sheet resistance/doping (reflectivity, XRD, REED, TEM, SIMS)
900	0.53–6.6	DCS	C_2_H_2_	-	64	(111),(220)	0.01	29	[26]	SiC_(s)_ (XRD) (C/Si)_(s)_~1 (XPS)	NH_3_ FR and P study on structure, growth rate, residual stress, and resistivity (4PP, XRD, AFM, XPS, AFT spectrometry)
1300–1500	1013	MTS	C_3_H_8_	1584	-	(111),(220),(311)	-	-	[81]	(XRD)-T SiC_(s)_ + Si_(s)_, T < 1500 °C SiC_(s)_,T > 1500 °C (XRD)-(C/Si)_(g)_ SiC_(s)_ + Si_(s)_, (C/Si)_(g)_: 1 SiC_(s)_, (C/Si)_(g)_: 2	T, P, and C_3_H_8_ FR study on structure and orientation. Investigation of hardness (XRD, SEM)
1350	40	SiCl_4_	CH_4_	1200	100–300	(111),(200),(220),(311)	0.1	-	[119]	SiC_(s)_ (XRD)	HLCVD: T and N_2_ FR study on growth rate, structure, surface morphology, doping, conductivity (XRD, FESEM, Raman, NIR, Hall, XPS)
950–1300	<1013	MTS	MTS	-	-	(111),(220),(311)	-	-(200–300)	[150]	SiC_(s)_ (XRD)	T, MTS FR, film thickness, and substrate thickness investigation of residual and intrinsic stress (XRD, TEM, Laser profilometry)
1000–1350	2	MTS	MTS	41,100	-	-	-	-	[77]	-	MOCVD: T study on growth rate and structure + surface morphology in a batch of graphite substrates (CFD and experiments: SEM, surface roughness)
1300	150	MTS	MTS	-	-	-	-	-	[151]	SiC_(s)_ (XRD)	MOCVD: T study on growth rate and structure + surface morphology with in a batch of graphite substrates (CFD and experiments: XRD, SEM, surface roughness, and thermal resistance)
800–900	0.6–6.6	DCS	C_2_H_2_	-	-	(111)	-	(−41)–695	[152]	SiC_(s)_ (XRD) (C/Si)_(s)_~1 (XPS)	T and P study on growth rate and structure. Chemical stability in KOH solution (XRD, SEM, XPS, AFM)
900–1070	20	TCS	C_3_H_8_	-	-	(111),(200),(220),(311)	-	-	[97]	SiC_(s)_ (XRD) SiC_(s)_ + C_(s)_ (Raman)	T and FR study on structure, preferential orientation, and stacking faults investigation of (XRD, Raman, TEM)
500–1150	20–100	VTC	VTC	100–500	-	-	-	-	[100]	SiC_(s)_ + C_(s)_ (EDX) SiC_(s)_ + C_(s)_ (Raman) C_(s)_ increases for T: 1000–1150 °C	T, P, and FR study on growth rate, structure, and chemical composition (FTIR, SEM, EDX, Raman, XRD, EPMA, TEM)

## Data Availability

Data are contained within the article.

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
