# Peer review of "Progress in Polycrystalline SiC Growth by Low Pressure Chemical Vapor Deposition and Material Characterization"

_micromachines, 2025, doi:10.3390/mi16030276_

Round 1
Reviewer 1 Report
Comments and Suggestions for Authors
The manuscript submitted for publication is devoted to reviewing the current state of research on the growth, structure, and properties of polycrystalline silicon carbide chemically vapor deposited at low pressure (LPCVD). This material is unique and widely sought after due to its mechanical, thermal, and electrical properties. In the review, the authors present a list of key achievements in this field of materials science based on the analysis of more than 180 sources over a time span of more than half a century. As a result of the analysis of the literature sources, the authors concluded that only the effect of temperature on the growth, structure, and properties of SiC LPCVD has been sufficiently investigated, while the effects of other conditions (pressure, chemical composition of the growth environment, etc.) remain not fully understood, highlighting the need for further research in these areas. The review is written clearly, is supplemented by a large number of illustrations, and will undoubtedly be useful for specialists in the field. Therefore, I recommend publishing the manuscript under consideration.
There are a few minor comments.
- Table 1: Please clarify data on ‘Intrinsic carrier concentration (cm-3)’, values are too low.
- Line 464: 0.25% - the percent sign is unnecessary.
- Line 632: Table 3 -> Table 4.
- Line 633: Table 4 -> Table 5.
- Line 968, 970: Please note that the orientation of the planes is indicated in parentheses, not square brackets.
- Line 1003: CTE values are sometimes given in ‘⁰C-1’ and sometimes in ‘K-1’. Does it make any difference? We need to be consistent.
Author Response
We attach here our word file, answering the questions and the remarks of the reviewer, in detail.

Reviewer 2 Report
Comments and Suggestions for Authors
This manuscript provides a comprehensive and well-structured review of the current technological trends in polycrystalline SiC growth using the LPCVD method. The paper effectively consolidates theoretical and experimental insights, offering valuable perspectives on this industrially significant technology. By summarizing key advancements in the field, this review serves as a useful resource for researchers and engineers working on SiC materials.
To further enhance the quality of the manuscript, I suggest the following comment to the abstract of this manuscript. The abstract provides a good overview of the paper, but it does not explicitly highlight why LPCVD is particularly important for polycrystalline SiC growth. Since LPCVD is a key focus of this review, it would be beneficial to briefly mention its advantages and significance in the abstract.
Comments on the Quality of English Language
I noticed some typo and grammatical errors throughout the manuscript. I recommend carefully proofreading the text to correct these issues to further improve readability and clarity.
Author Response
"Please see the attachment."
Please consult the attached word file, in which we answer the questions of the reviewer in detail.

Reviewer 3 Report
Comments and Suggestions for Authors
The authors give a comprehensive overview of the synthesis of SiC with LPCVD processes. The aim of the review is a deeper understanding of the dependencies of process parameters with regard to the synthesis itself and the properties of SiC. The authors did an extensive paper work and I would suggest publishing after considering some minor comments
The authors stated in the introduction: “The motivation for this review stems from the increasing demand for high-performance SiC material and the need for a deeper understanding of the LPCVD growth process and its impact on material properties” and in 2.2. the authors show different application scenarios. How do the authors interpret high performance SiC? Maybe it could be helpful if the authors not only said that for instance, good conductivity is necessary, but also backed this up with numerical values. This would also help the reader during the extensive study to not lost focus.
The authors could be consistent in naming SiC or silicon carbide, e.g. in 2.1.2 till 2.1.5.
3.1: Theoretical work: The authors show two strategies to simulate the LP-CVD process: In your opinion, which simulation is better suited to mapping the process? A short paragraph comparing both might be helpful.
Author Response

(The authors gave the same response as above.)
